# A Schrödinger Eigenfunction Method for Long-Horizon Stochastic Optimal Control

**Louis Claeys**
ETH Zürich,
Department of Mathematics
louis.claeys@live.be

**Artur Goldman**
ETH Zürich, ETH AI Center
Institute for Machine Learning
artur.goldman@ai.ethz.ch

**Zebang Shen**
ETH Zürich,
Institute for Machine Learning
zebang.shen@inf.ethz.ch

**Niao He**
ETH Zürich,
Institute for Machine Learning,
niao.he@inf.ethz.ch

## Abstract

High-dimensional stochastic optimal control (SOC) becomes harder with longer planning horizons: existing methods scale linearly in the horizon $T$, with performance often deteriorating exponentially. We overcome these limitations for a subclass of linearly-solvable SOC problems—those whose uncontrolled drift is the gradient of a potential. In this setting, the Hamilton-Jacobi-Bellman equation reduces to a linear PDE governed by an operator $\mathcal{L}$. We prove that, under the gradient drift assumption, $\mathcal{L}$ is unitarily equivalent to a Schrödinger operator $\mathcal{S} = -\Delta + \mathcal{V}$ with purely discrete spectrum, allowing the long-horizon control to be efficiently described via the eigensystem of $\mathcal{L}$. This connection provides two key results: first, for a symmetric linear-quadratic regulator (LQR), $\mathcal{S}$ matches the Hamiltonian of a quantum harmonic oscillator, whose closed-form eigensystem yields an analytic solution to the symmetric LQR with *arbitrary* terminal cost. Second, in a more general setting, we learn the eigensystem of $\mathcal{L}$ using neural networks. We identify implicit reweighting issues with existing eigenfunction learning losses that degrade performance in control tasks, and propose a novel loss function to mitigate this. We evaluate our method on several long-horizon benchmarks, achieving an order-of-magnitude improvement in control accuracy compared to state-of-the-art methods, while reducing memory usage and runtime complexity from $\mathcal{O}(Td)$ to $\mathcal{O}(d)$.

## 1 Introduction

Stochastic optimal control (SOC) concerns the problem of directing a stochastic system, typically modeled by a stochastic differential equation (SDE), to minimize an expected total cost. SOC has found applications in various domains, e.g. stochastic filtering (Mitter, 1996), rare event simulation in molecular dynamics (Hartmann & Schütte, 2012; Hartmann et al., 2014), robotics (Gorodetsky et al., 2018) and finance (Pham, 2009).

Built on the principle of dynamic programming, the global optimality condition of SOC can be expressed by the Hamilton-Jacobi-Bellman (HJB) equation. In this paper, we focus on the *affine control* setting commonly considered in the literature (Fleming & Rishel, 1975; Kappen, 2005b; Fleming & Soner, 2006; Yong & Zhou, 1999; Domingo-Enrich et al., 2024b; Nüsken & Richter, 2021; Carius et al., 2022; Holdijk et al., 2023), where the control affects the state of the system linearly. This setting is of interest since the optimal control will exactly match the gradient of the value function of SOC problem and hence the corresponding HJB equation can be drastically simplified.

A canonical special case of this affine-control framework is the linear–quadratic regulator (LQR), in which the uncontrolled dynamics follow an Ornstein–Uhlenbeck linear SDE and both the running cost and the terminal cost are quadratic. One can show that in the LQR setting the value function retains a quadratic form—indeed, it is quadratic at terminal time because the terminal cost is quadratic—and

that the optimal feedback control is linear w.r.t. the state. Consequently, the associated SOC problem admits an explicit solution via the finite-dimensional matrix Riccati differential equation (van Handel, 2007)

To obtain the optimal control in more general settings requires numerical procedures. For low-dimensional problems, classical grid-based PDE solvers may be used, but these suffer from the curse of dimensionality. This has led to several works proposing the use of neural networks (NN) to solve the HJB equation in more complex high-dimensional settings, either through a forward-backward stochastic differential equation (FBSDE) approach (Han et al., 2018; Ji et al., 2022; Andersson et al., 2023; Beck et al., 2019) or so-called iterative diffusion optimization (IDO) methods, which sample controlled trajectories through simulation and update the NN parameter using stochastic gradients from automatic differentiation (Nüsken & Richter, 2021; Domingo-Enrich et al., 2024b;a). A more comprehensive review on existing methods for short-horizon SOC can be found in Appendix C.

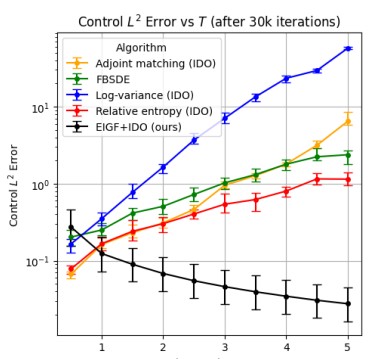

While these methods have proven successful, their performance suffers as the time horizon $T$ grows. Both the memory requirement and per-iteration runtime increase at least linearly in $T$. Additionally, it holds that error estimates for the deep FBSDE method worsen as $T$ increases (Han & Long, 2020, Theorem 4), and for IDO methods using importance sampling the weight variance may increase exponentially in $T$ (Liu et al., 2018). These limitations were observed empirically in Assabumrungrat et al. (2024); Domingo-Enrich et al. (2024b), and were reproduced in our experiments (see Figure 1).

Figure 1: Performance degradation as time horizon $T$ increases for different methods (see Appendix E for details).

**Linearly-solvable HJB.** The HJB equation is in general nonlinear. However, in the special case where the system's diffusion coefficient matches the affine-control mapping, the Cole–Hopf transform can eliminate the nonlinearity (Evans, 2022). Specifically, let $V(x,t)$ denote the value function of the SOC problem and define a new function $\psi := \exp(-V)$. Under this transformation, the HJB is equivalently rewritten as the following *linear PDE*

$$\partial_t \psi(x,t) = \mathcal{L}\psi(x,t), \quad \psi(x,T) = \psi_T(x) \tag{1}$$

for some *linear* operator $\mathcal{L}$. Moreover, the optimal control $u$, which exactly matches the gradient of the value function $-\partial_x V$, can be obtained as $u^* = \partial_x \log \psi$ (Kappen, 2005b).

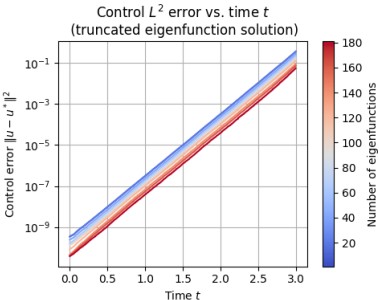

Working with a linear PDE brings several clear benefits over a nonlinear one, such as a simplified analysis — questions of well-posedness and solution regularity of the PDE become much more tractable — but more importantly algorithmic insight: One can borrow ideas from the finite-dimensional linear ODE $\dot{u}(t) + A u(t) = 0$, where $A$ is a real symmetric matrix. This ODE has a closed-form solution $u(t) = e^{-At} u(0)$, and because the matrix exponential $e^{-At}$ acts simply on $A$'s eigenvalues, expanding $u(0)$ in the corresponding eigenvectors yields an efficient numerical scheme.

Figure 2: Diminishing returns from increasing the number of eigenfunctions for an LQR in $d = 20$ dimensions.

While the domain of the operator $\mathcal{L}$ is infinite-dimensional (acting on functions rather than finite vectors), the same exponential-integrator principle applies (Theorem 1), and we can write $\psi_t = e^{(t-T)\mathcal{L}}\psi_T$, where $\left(e^{s\mathcal{L}}\right)_{s\geq 0}$ is a semi-group (Renardy & Rogers, 2006, Chapter 12). Of course, carrying it out in an infinite-dimensional setting introduces additional technical challenges that must be carefully addressed. In particular, expanding $\psi_T$ as a series of eigenfunctions requires $\mathcal{L}$ to possess a *discrete spectrum*, i.e. one can find an orthonormal basis of eigenfunctions $(\phi_i)_{i\in\mathbb{N}}$ and corresponding eigenvalues $\lambda_0 < \lambda_1 \leq \cdots$ such that $\mathcal{L}\phi_i = \lambda_i\phi_i$ for all $i \in \mathbb{N}$. Under this assumption, the optimal control can be informally written as

$$u^*(x,t) = \partial_x \log \phi_0(x) + \mathcal{O}\left(e^{-(\lambda_1-\lambda_0)(T-t)}\right) \quad \text{for any fixed } t \text{ as } T \to \infty, \tag{2}$$

where $\mathcal{O}$ hides eigenfunctions $\phi_i$ with $i \geq 1$. A precise statement can be found in Theorem 3. To turn the above formula into a practical, long-horizon SOC algorithm, we need (1) **Spectral verification.** Prove that $\mathcal{L}$ indeed has a discrete spectrum; (2) **Eigenfunction identification.** Compute the spatial derivative of the principal eigenfunction $\phi_0$.

**Our approach: Reduction to Schrödinger operator.** In this paper, to guarantee the spectral verification, we assume that the drift of the dynamics is *the gradient of a potential*. Such problems of controlling a diffusion process with gradient drift show up in overdamped molecular dynamics (Schütte et al., 2012), mean-field games (Bakshi et al., 2019; Grover et al., 2018), the control of particles with interaction potentials (Carrillo et al., 2020; Totzeck & Pinnau, 2020), as well as social models for opinion formation (Castellano et al., 2009; Albi et al., 2017).

In this setting, the operator $\mathcal{L}$ in (1) is unitarily equivalent to the Schrödinger operator $\tilde{\mathcal{L}} = -\Delta + \mathcal{V}$, where $\Delta$ is the Laplacian and $\mathcal{V}$ is an effective potential determined by the original drift and running cost. Because $\tilde{\mathcal{L}}$ is known to have a purely discrete spectrum on $L^2$ and unitary equivalence preserves the spectral properties, the operator $\mathcal{L}$ in (1) likewise enjoys a discrete spectrum.

This result forms the basis of our new framework for long-horizon SOC, where the problem is reformulated as learning the eigensystem of a Schrödinger operator. Indeed, (2) shows that the top eigenfunction $\phi_0$ determines the long-term control, with corrections decaying exponentially with rate $\lambda_1 - \lambda_0$. We address the problem of eigenfunction identification in the following two scenarios:

- **Closed-form solution for LQR with non-quadratic terminal cost.** When the drift is linear with a *symmetric* coefficient matrix and the running cost is quadratic in the state, the resulting Schrödinger operator $\tilde{\mathcal{L}}$ coincides with the Hamiltonian of the quantum harmonic oscillator. Its eigenvalues and eigenfunctions are known explicitly (see Lemma 4 or Griffiths & Schroeter (2018)), i.e. $\{\lambda_i, \phi_i\}_{i \in \mathbb{N}}$ are available in closed form. Consequently, our framework yields a fully explicit expression for the corresponding SOC. This removes the requirement of quadratic terminal cost in the the classical LQR solution.

- **Neural network-based approach for general gradient drift.** For general gradient–drift dynamics, we introduce a hybrid neural-network method to approximate the optimal control efficiently: Rather than attempting to learn the full spectrum of $\mathcal{L}$—which is prohibitively expensive and yields rapidly diminishing returns (fig. 2)—we exploit the exponential decay of the higher modes w.r.t. $T - t$ in eq. (2). Concretely, whenever $T - t$ exceeds a modest threshold (in our experiments $T - t \geq 1$), it suffices to approximate the control using only the top eigenfunction $\phi_0$. For the remaining period ($t$ very close to $T$), we switch to established FBSDE/IDO solvers to handle the short-horizon SOC.

  We propose a novel deep learning strategy for the task of learning the eigenfunction, tailored to SOC. While such a task has been extensively studied in the literature, previous approaches either optimize a variational Ritz objective (E & Yu, 2018; Zhang et al., 2022; Cabannes & Bach, 2024) or minimize the residual norm $\|\mathcal{L}\psi - \lambda\psi\|^2$ (Jin et al., 2022; Zhang et al., 2024; Nusken & Richter, 2023). However, these losses implicitly reweight spatial regions, causing them to fail to learn the control in regions where the value function $V$ is large—the most crucial areas. To eliminate this bias, we introduce a *relative* eigenfunction loss, $\|\mathcal{L}\psi/\psi - \lambda\|^2$, which removes the undesired weighting and robustly recovers the dominant eigenpair needed for control synthesis.

Our contributions are summarized as follows:

- We provide a new perspective on finite-horizon gradient-drift SOC problems, linking their solution to the eigensystem of a Schrödinger operator (Theorem 3). This yields a previously unreported closed-form solution to the symmetric LQR with arbitrary terminal cost (Theorem 4).

- With this framework, we introduce a new procedure for solving SOC problems over long horizons by learning the operator's eigenfunctions with neural networks. We show that existing eigenfunction solvers can be ill-suited for this task due to an implicit reweighting in the used loss, and propose a new loss function to remedy this.

- We perform experiments in different settings to evaluate the proposed method against state-of-the-art SOC solvers, showing roughly an order of magnitude improvement in control $L^2$ error on several high-dimensional ($d = 20$) long-horizon problems.

## 2 PRELIMINARIES

### 2.1 STOCHASTIC OPTIMAL CONTROL

Fix a filtered probability space $(\Omega, \mathcal{F}, (\mathcal{F}_t)_{t \geq 0}, \mathbb{P})$, and denote $(W_t)_{t \geq 0}$ a Brownian motion in this space. Let $(X_t^u)_{t \geq 0}$ denote the random variable taking values in $\mathbb{R}^d$ defined through the SDE

$$\mathrm{d}X_t^u = (b(X_t^u) + \sigma u(X_t^u, t))\,\mathrm{d}t + \sqrt{\beta^{-1}}\sigma \mathrm{d}W_t, \qquad X_0^u \sim p_0 \tag{3}$$

where $u : \mathbb{R}^d \times [0, T] \to \mathbb{R}^d$ is the control, $b : \mathbb{R}^d \to \mathbb{R}^d$ is the base drift, $\sigma \in \mathbb{R}^{d \times d}$ is the diffusion coefficient (assumed invertible) and $\beta \in \mathbb{R}_0^+$ is an inverse temperature characterizing the noise level. Note that we assume the drift and noise to be time-independent, in contrast to Nüsken & Richter (2021); Domingo-Enrich et al. (2024b). Under some regularity conditions on the coefficients and control $u$ described in Appendix A, the SDE (3) has a unique strong solution. In stochastic optimal control, we view the dynamics $(b, \sigma, \beta)$ as given and consider the problem of finding a control $u$ which minimizes the cost functional

$$J(u; x, t) = \mathbb{E}\left[\int_t^T \left(\frac{1}{2}\|u(X_t^u, t)\|^2 + f(X_t^u)\right)\mathrm{d}t + g(X_T^u)\,\middle|\, X_t = x\right] \tag{4}$$

where $f : \mathbb{R}^d \to \mathbb{R}$ denotes the running cost and $g : \mathbb{R}^d \to \mathbb{R}$ denotes the terminal cost. We denote this optimal control as $u^*(x, t) = \arg\min_{u \in \mathcal{U}} J(u; x, t)$, with $\mathcal{U}$ the set of admissible controls. To analyze this problem, one defines the *value function $V$*, which is defined as the minimum achievable cost when starting from $x$ at time $t$,

$$V(x, t) := \inf_{u \in \mathcal{U}} J(u; x, t). \tag{5}$$

In this case, the optimal control $u^*$ that minimizes the objective (4) is obtained from the value function through the relation $u^* = -\sigma^T \nabla V$, as described in (Nüsken & Richter, 2021, Theorem 2.2).

**Hamilton-Jacobi-Bellman equation.** A well-known fundamental result is that when the value function $V$ is sufficiently regular, it satisfies the following partial differential equation, called the Hamilton-Jacobi-Bellman (HJB) equation (Fleming & Rishel, 1975):

$$\partial_t V + \mathcal{K}V = 0 \quad \text{in } \mathbb{R}^d \times [0, T], \quad V(\cdot, T) = g \quad \text{on } \mathbb{R}^d, \tag{6}$$

$$\text{where } \mathcal{K}V = \frac{1}{2\beta}\mathsf{Tr}(\sigma\sigma^T \nabla^2 V) + b^T \nabla V - \frac{1}{2}\|\sigma^T \nabla V\|^2 + f. \tag{7}$$

The so-called *verification theorem* states (in some sense) the converse: if a function $V$ satisfying the above PDE is sufficiently regular, it coincides with the value function (5) corresponding to (3)-(4), see (Fleming & Rishel, 1975, Section VI.4) and (Pavliotis, 2014, Sec. 2.3).

**A linear PDE reformulation** Although the HJB equation (6) is nonlinear in general, it was shown in Kappen (2005b) that for a specific class of optimal control problems (which includes the formulation (3)-(4)), a suitable transformation allows for a linear reformulation of (6). More specifically, when parametrizing $V(x, t) = -\beta^{-1} \log \psi(x, \frac{1}{2\beta}(T - t))$, the nonlinear terms cancel, and (6) becomes

$$\begin{cases} \partial_\tau \psi + \mathcal{L}\psi = 0, \\ \psi(\cdot, 0) = \psi_0, \end{cases} \quad \text{where } \mathcal{L}\psi = -\mathsf{Tr}(\sigma\sigma^T \nabla^2 \psi) - 2\beta b^T \nabla \psi + 2\beta^2 f \cdot \psi, \quad \psi_0 = \exp(-\beta g), \tag{8}$$

and we have introduced the variable $\tau = (2\beta)^{-1}(T - t)$. This is precisely the abstract form (1), but with a time reversal. For more details on this result, we refer to Appendix B. To simplify the presentation, we will often assume w.l.o.g. that $\sigma = I$ (see Appendix A), so that $\mathsf{Tr}(\sigma\sigma^T \nabla^2) = \Delta$.

### 2.2 EIGENFUNCTION SOLUTIONS

Consider a Hilbert space $\mathcal{H}$ with inner product $\langle \cdot, \cdot \rangle$, and a linear operator $\mathcal{L} : D(\mathcal{L}) \to \mathcal{H}$ defined on a dense subspace $D(\mathcal{L}) \subset \mathcal{H}$. Then we have the following standard definition:

**Definition 1** *An element $\phi \in \mathcal{H}$ with $\phi \neq 0$ is an* eigenfunction *of $\mathcal{L}$ if there exists a $\lambda \in \mathbb{C}$ such that $\mathcal{L}\phi = \lambda\phi$. The value $\lambda$ is called an* eigenvalue *of $\mathcal{L}$ (corresponding to $\phi$), and the dimension of the null space of $\mathcal{L} - \lambda I$ is called the* multiplicity *of $\lambda$.*

In a finite-dimensional setting, the study of a linear operator $A$ is drastically simplified when we have access to an orthonormal basis of eigenvectors. Similarly, some operators admit a countable set of eigenfunctions $(\phi_i)_{i \in \mathbb{N}}$ which forms an orthonormal basis of the Hilbert space $\mathcal{H}$. When such an eigensystem exists, the following theorem proven in Appendix B gives a solution to the PDE (8) in terms of the eigensystem. A rigorous connection between this solution to (8) and solutions to (6) is explored in Appendix B.

**Theorem 1 (Restatement of Theorem VIII.7 in (Reed & Simon, 1980))** *Let $\mathcal{L}$ be an essentially self-adjoint[1], densely defined operator on $\mathcal{H}$ which admits an orthonormal basis of eigenfunctions $(\phi_i, \lambda_i)_{i \in \mathbb{N}}$. Assume further that the $\lambda_i$ are bounded from below (write $\lambda_0 \leq \lambda_1 \leq \ldots$) and do not have a finite accumulation point. Then a solution to the abstract evolution problem in* (8) *is given by*

$$\psi(\tau) = \sum_{i \in \mathbb{N}} e^{-\lambda_i \tau} \langle \phi_i, \psi_0 \rangle \phi_i. \tag{9}$$

While we originally aimed to solve the HJB equation (6), in this work, we solve the linear PDE (8) as surrogate. In Section B.7, we show that under standard growth and regularity assumptions, the above semigroup solution coincides with the unique viscosity solution of the original HJB equation, up to an inverse Hope-Cole transformation. See Theorem 10 for a formal statement.

## 3 OUR FRAMEWORK

### 3.1 SPECTRAL PROPERTIES OF THE SCHRÖDINGER OPERATOR

In order to apply Theorem 1, we must establish conditions under which the operator $\mathcal{L}$ in (8) satisfies the required properties. In order for $\mathcal{L}$ to be symmetric, we assume

**(A1)** The drift $b$ in (3) is described by a gradient field: $b(x) = -\nabla E(x)$.

Define the measure $\mu$ on $\mathbb{R}^d$ with density $\mu(x) = \exp(-2\beta E(x))$, and consider the weighted Lebesgue space $L^2(\mu)$. Note that we do *not* require $\mu$ to be a finite measure. Under the assumption **(A1)**, the operator appearing in (8) becomes the following operator on $L^2(\mu)$:

$$\mathcal{L} : D(\mathcal{L}) \subset L^2(\mu) \to L^2(\mu) : \psi \mapsto \mathcal{L}\psi = -\Delta\psi + 2\beta\langle\nabla E, \nabla\psi\rangle + 2\beta^2 f\psi. \tag{10}$$

Under mild regularity conditions, we can further show that $\mathcal{L}$ is essentially self-adjoint (Appendix B). Furthermore, it can be shown (Appendix B) that

$$U\mathcal{L}U^{-1} = -\Delta + \beta^2\|\nabla E\|^2 - \beta\Delta E + 2\beta^2 f \tag{11}$$

where $U : L^2(\mu) \to L^2(\mathbb{R}^d) : \psi \mapsto e^{-\beta E}\psi$ is a unitary operator, so that $\mathcal{L}$ is unitarily equivalent to the well-known Schrödinger operator $\mathcal{S} = -\Delta + \mathcal{V}$ on $L^2(\mathbb{R}^d)$, which forms the cornerstone of the mathematical formulation of quantum physics. Its properties have been studied to great extent (Reed & Simon, 1978), allowing us to invoke well-known results on the properties of the Schrödinger operator to study the behavior of $\mathcal{L}$. In particular, the following assumption on $E$ and $f$ is enough to guarantee that $\mathcal{L}$ satisfies all the desired properties (see Appendix B).

**(A2)** For the energy $E$ and running cost $f$, $\mathcal{V} := \beta\|\nabla E\|^2 - \Delta E + 2\beta f$ satisfies $\mathcal{V} \in L^2_{loc}(\mathbb{R}^d)$, $\exists C \in \mathbb{R}, \forall x \in \mathbb{R}^d : C \leq \mathcal{V}(x)$, and $\mathcal{V}(x) \to \infty$ as $\|x\| \to \infty$.

**Theorem 2 (Restatement of (Reed & Simon, 1978), Theorem XIII.67, XIII.64, XIII.47)** *Suppose (A2) is satisfied. Then the operator $\mathcal{L}$ in* (10) *is densely defined and essentially self-adjoint. Moreover, it admits a countable, orthonormal basis of eigenfunctions. In addition, the eigenvalues are bounded from below and do not have a finite accumulation point, the lowest eigenvalue $\lambda_0$ has multiplicity one, and the associated eigenfunction (called the* ground state*, or in our context the* top eigenfunction*) can be chosen to be strictly positive.*

**Remark 1** *Previous studies have linked the Schrödinger operator to optimal control in contexts distinct from ours: for example, Schütte et al. (2012) and Bakshi et al. (2020) analyze the stationary HJB equation $\mathcal{K}V_\infty = \lambda$ (see Remark 2), whereas Kalise et al. (2025) explores distribution-level control of the Fokker–Planck equation.*

---

[1] An operator is called essentially self-adjoint if its closure is self-adjoint. See Reed & Simon (1980) and Reed & Simon (1975) for more details.

### 3.2 EIGENFUNCTION CONTROL

From the previous discussion, we obtain the following result, which links the eigenfunctions of $\mathcal{L}$ with the optimal control problem.

**Theorem 3** *Suppose (A1)-(A2) are satisfied. Then $\mathcal{L}$ satisfies all assumptions in Theorem 1, hence the solution of the optimal control problem* (3)-(4) *is given by*

$$u^*(x,t) = \beta^{-1}\left(\nabla \log \phi_0(x) + \nabla \log\left(1 + \sum_{i>0} \frac{\langle e^{-\beta g}, \phi_i\rangle_\mu}{\langle e^{-\beta g}, \phi_0\rangle_\mu} e^{-\frac{1}{2\beta}(\lambda_i - \lambda_0)(T-t)} \frac{\phi_i(x)}{\phi_0(x)}\right)\right). \quad (12)$$

*where $(\phi_i, \lambda_i)_{i\in\mathbb{N}}$ is the orthonormal eigensystem of $\mathcal{L}$ defined in* (10)*, and $\lambda_0 < \lambda_1 \leq \ldots$.*

Thus, the long-term optimal control ($t \ll T$) is given by $\beta^{-1}\nabla \log \phi_0$, and the corrections decay exponentially, motivating truncation of the series in (12).

### 3.3 CLOSED-FORM SOLUTION FOR THE SYMMETRIC LQR

When $E$ and $f$ are quadratic, the Schrödinger operator associated with the optimal control system is the Hamiltonian for the harmonic oscillator, which has an exact solution (Appendix B):

**Theorem 4** *Suppose $b(x) = -Ax$ for some symmetric matrix $A \in \mathbb{R}^{d\times d}$, and $f(x) = x^T P x$ for some matrix $P \in \mathbb{R}^{d\times d}$ such that $A^T A + 2P$ is positive definite, and has diagonalization $U^T \Lambda U$. Then the orthonormal eigensystem of the operator $\mathcal{L}$ given in* (10) *is described through*

$$\phi_\alpha(x) = \frac{\exp\left(-\frac{\beta}{2}x^T\left(-A + U^T\Lambda^{1/2}U\right)x\right)}{(\lambda\pi)^{d/4}}\prod_{i=1}^d \frac{\Lambda_{ii}^{1/8}}{\sqrt{2^{\alpha_i}(\alpha_i!)}}H_{\alpha_i}\left(\sqrt{\beta}(\Lambda^{1/4}Ux)_i\right), \quad (13)$$

$$\lambda_\alpha = \beta\left(-\mathsf{Tr}(A) + \sum_{i=1}^d \Lambda_{ii}^{1/2}(2\alpha_i + 1)\right). \quad (14)$$

*where $\alpha \in \mathbb{N}^d$ and $H_i$ denotes the $i$th physicist's Hermite polynomial. We can bijectively map $\alpha \in \mathbb{N}^d$ to $i \in \mathbb{N}$ by ordering the eigenvalues* (14)*, yielding the same representation as before.*

Combined with Theorem 3, this yields a closed-form solution for the optimal control problem with symmetric linear drift, quadratic running cost and arbitrary terminal reward.

## 4 NUMERICAL METHODS

We propose a hybrid method with two components: Far from the terminal time $T$, we learn the top eigenfunction $\phi_0$ and simply use $\partial_x \log \phi_0$ as the control (c.f. eq. (2)); Close to the terminal time, e.g. $t \geq T - 1$, we use an existing solver to learn an additive short-horizon correction to the control.

### 4.1 LEARNING EIGENFUNCTIONS

In absence of a closed-form solution, a wide range of numerical techniques exist for solving the eigenvalue problem for the operator $\mathcal{L}$ in (10). Classically, the eigenfunction equation is projected onto a finite-dimensional subspace to yield a Galerkin/finite element method, see Chaitin-Chatelin (1983). In high dimensions, these methods often perform poorly, motivating deep learning approaches which differ from each other mainly in the loss function used. We will only discuss methods for learning a single eigenfunction, referring to Appendix C for extensions to multiple eigenfunctions. An overview of the deep leraning algorithm for learning eigenfunctions is given in Algorithm 1 in Appendix E.

**PINN loss** Based on the success of physics-informed neural networks (PINNs) (Raissi et al., 2019), one idea is to design a loss function that attempts to enforce the equation $\mathcal{L}\phi = \lambda\phi$ via an $L^2$ loss, as done in Jin et al. (2022). The idea is to consider some density $\rho$ on $\mathbb{R}^d$, and define the loss function

$$\mathcal{R}^\rho_{\text{PINN}}(\phi) = \|\mathcal{L}[\phi] - \lambda\phi\|_\rho^2 + \alpha\mathcal{R}^\rho_{reg}(\phi), \quad \mathcal{R}^\rho_{reg}(\phi) = (\|\phi\|_\rho^2 - 1)^2 \quad (15)$$

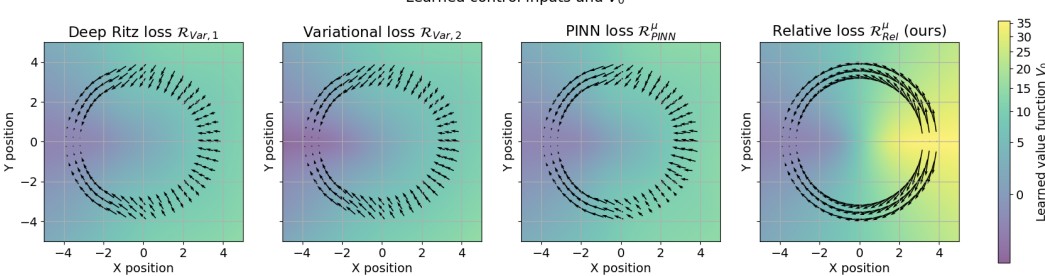

Figure 3: Learned controls (arrows) and $V_0$ for different eigenfunction losses. Existing methods fail to learn the correct control in regions where $V_0$ is large due to implicit reweighting.

where $\alpha > 0$ is a regularizer to avoid the trivial solution $\phi = 0$. The eigenvalue $\lambda$ is typically also modeled as a trainable parameter of the model, or obtained through other estimation procedures.

**Variational loss** A second class of loss functions is based on the variational characterization of the eigensystem of $\mathcal{L}$. Since $\mathcal{L}$ is essentially self-adjoint with orthogonal eigenbasis in a subset of $L^2(\mu)$ ($\mu$ is defined below **(A1)**), it holds that (see (Reed & Simon, 1978, Theorem XIII.1))

$$\lambda_0 = \inf_{\psi \in L^2(\mu)} \frac{\langle \psi, \mathcal{L}\psi \rangle_\mu}{\langle \psi, \psi \rangle_\mu} \tag{16}$$

where the infimum is obtained when $\mathcal{L}\psi = \lambda_0 \psi$. This motivates the loss functions

$$\mathcal{R}_{\mathrm{Var},1}(\phi) = \frac{\langle \phi, \mathcal{L}\phi \rangle_\mu}{\langle \phi, \phi \rangle_\mu} + \alpha \mathcal{R}_{reg}^\mu(\phi), \quad \mathcal{R}_{\mathrm{Var},2}(\phi) = \langle \phi, \mathcal{L}\phi \rangle_\mu + \alpha \mathcal{R}_{reg}^\mu(\phi). \tag{17}$$

The first of these, sometimes called the *deep Ritz loss*, was introduced in E & Yu (2018), and the second was described in Cabannes & Bach (2024); Zhang et al. (2022). These loss functions do not require prior knowledge of the eigenvalue $\lambda_0$.

### 4.1.1 IMPLICIT REWEIGHTING IN PREVIOUS APPROACHES

The exponential decay of the correction term in eq. (2) suggests that the optimal control $u^*$ can be approximated by the spatial derivative of *logarithm* of the top eigenfunction. We therefore parameterize in our implementation $\phi = \exp(-\beta V_0)$, where $V_0$ is some neural network and $\beta$ is the temperature constant. This choice also enforces the *strict positivity* of $\phi$, which matches the same property of $\phi_0$ established in Theorem 2.

Adopting such a parameterization, for the PINN and variational losses, it holds that

$$\mathcal{R}_{\mathrm{PINN}}^\rho(e^{-\beta V_0}) = 4\beta^4 \left\| e^{-\beta V_0} \left( \mathcal{K}V_0 - \frac{\lambda_0}{2\beta^2} \right) \right\|_\rho^2 + \alpha \mathcal{R}_{reg}^\rho(e^{-\beta V_0}) \tag{18}$$

$$\mathcal{R}_{\mathrm{Var},2}(e^{-\beta V_0}) = 2\beta^2 \int e^{-2\beta V_0} \mathcal{K}V_0 \, \mathrm{d}\mu + \alpha \mathcal{R}_{reg}^\mu(e^{-\beta V_0}) \tag{19}$$

where $\mathcal{K}$ is the HJB operator (7). Because both (18) and (19) incorporate an exponential factor that vanishes where $V_0$ is large, these losses become effectively blind to errors in high-$V_0$ regions and are only able to learn where $V_0$ is small. This pathology is illustrated in Figure 3: Consider a 2D RING energy landscape $E$, whose minimizers lie on a circle, and a cost $f$ that grows linearly with the $x$-coordinate (see the full setup in Section E). The true optimal control remains tangential to the circle. In contrast, the controls obtained via the PINN and variational eigenfunction losses collapse in regions of large $V_0$, deviating sharply from the expected direction.

### 4.1.2 OUR APPROACH: REMOVING IMPLICIT REWEIGHTING VIA RELATIVE LOSS

Based on this observation, we propose to modify (15) to

$$\mathcal{R}_{Rel}^\rho(\phi) = \left\| \frac{\mathcal{L}\phi}{\phi} - \lambda \right\|_\rho^2 + \alpha \mathcal{R}_{reg}^\rho(\phi). \tag{20}$$

This loss function, which we call the *relative loss*, eliminates the implicit reweighting of the stationary HJB equation. Indeed, the same computation as before yields

$$\mathcal{R}_{Rel}^{\rho}(e^{-\beta V_0}) = 4\beta^4 \left\| \mathcal{K}V_0 - \frac{\lambda_0}{2\beta^2} \right\|_{\rho}^2 + \alpha \mathcal{R}_{reg}^{\rho}(e^{-\beta V_0}). \tag{21}$$

As a result, the relative loss remains sensitive even in regions where $\phi = e^{-\beta V_0}$ becomes small. This can also be empirically observed in the RING task, as illustrated in Figure 3. Instead of learning $V_0$ and $\lambda_0$ jointly, a natural idea is to combine the benefits of the above loss functions by first training with a variational loss (17) to obtain an estimate for the eigenvalue $\lambda_0$ and a good initialization of $V_0$, and then 'fine-tune' using (20). In practice, we also observed that this initialization is necessary for the relative loss (20) to converge.

**Remark 2** *We note that there is an alternative interpretation of* (21) *based on a separate class of control problems in which there is no terminal cost g, and an infinite-horizon cost is minimized, yielding a* stationary *or* ergodic *optimal control (Kushner, 1978). In this setting, $\mathrm{u}$ is related to a time-independent value function $V_{\infty}$ which satisfies a stationary HJB equation of the form $\mathcal{K}V_{\infty} = \lambda$. In low dimensions, these problems are solved through classical techniques such as basis expansions or grid-based methods, with no involvement of neural networks (Todorov, 2009).*

## 4.2 OUR HYBRID METHOD: COMBINING EIGENFUNCTIONS AND SHORT-HORIZON SOLVERS

For both IDO and FBSDE methods, every iteration requires the numerical simulation of an SDE, yielding a linear increase in computation cost with the time horizon $T$. We propose to leverage the eigenfunction solution given in Theorem 3 in order to scale these methods to longer time horizons as follows: first, parametrize the top eigenfunction as $\phi_0^{\theta_0} = \exp(-\beta V_0^{\theta_0})$ for a neural network $V_0^{\theta_0}$, and learn the parameters $\theta_0$ using the relative loss, as well as the first two eigenvalues $\lambda_0, \lambda_1$ (see Appendix E). Next, choose some cutoff time $T_{cut} < T$ and parametrize the control as

$$u_\theta(x,t) = \begin{cases} \beta^{-1}\nabla \log \phi_0^{\theta_0} & 0 \le t \le T_{cut}, \\ \beta^{-1}\left( \nabla \log \phi_0^{\theta_0}(x) + e^{-\frac{1}{2\beta}(\lambda_1-\lambda_0)(T-t)}v^{\theta_1}(x,t) \right) & T_{cut} < t \le T. \end{cases} \tag{22}$$

This control can then be used in an IDO/FBSDE algorithm to optimize the parameters $\theta_1$ of the additive correction $v^{\theta_1}$, a second neural network, near the terminal time. Crucially, this only requires simulation of the system in the interval $[T_{cut}, T]$, significantly reducing the overall computational burden and reducing the time complexity of the algorithm from $\mathcal{O}(Td)$ to $\mathcal{O}(d)$.

**Choice of $T_{cut}$.** Increasing $T_{\text{cut}} \to T$ raises computational cost, whereas choosing $T_{\text{cut}}$ too small degrades performance; this reflects the fundamental trade-off in selecting this parameter. One heuristic for choosing an appropriate order of magnitude is to observe that we would like the correction term given by the infinite series in Eq. (12) to be small for all $t \le T_{\text{cut}}$. Even without access to the inner products or eigenfunctions, we may consider the approximation

$$\exp\left(-\tfrac{1}{2\beta}(\lambda_1 - \lambda_0)(T - T_{\text{cut}})\right) = \varepsilon,$$

which yields

$$T - T_{\text{cut}} = -\frac{2\beta}{\lambda_1 - \lambda_0} \log \varepsilon.$$

Since we obtain empirical estimates of $\lambda_0$ and $\lambda_1$, this expression provides a practical guideline for determining the scale of $T_{\text{cut}}$, and thus can be implemented in practice.

## 5 EXPERIMENTS

To evaluate the benefits of the proposed method, we consider four different settings, QUADRATIC (ISOTROPIC), QUADRATIC (ANISOTROPIC), DOUBLE WELL, and RING. An additional setting QUADRATIC (REPULSIVE) with nonconfining energy is discussed in Appendix E. The first three are high-dimensional benchmark problems adapted from Nüsken & Richter (2021), modified to be long-horizon problems, where a ground truth can be obtained. Detailed information on the experimental setups, including computational costs, is given in Appendix E.

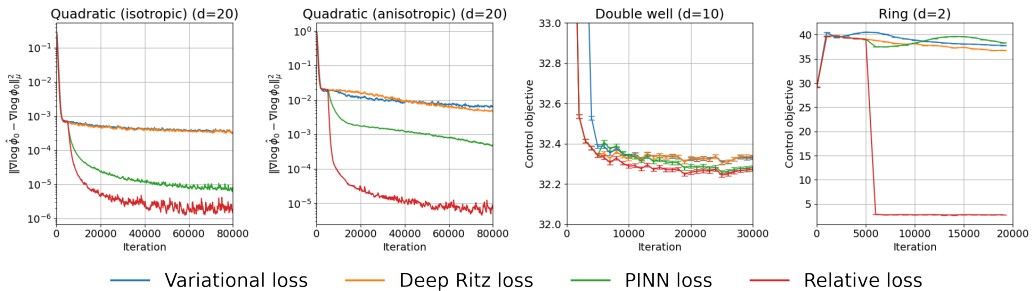

Figure 4: Comparison of the different eigenfunction losses (EMA).

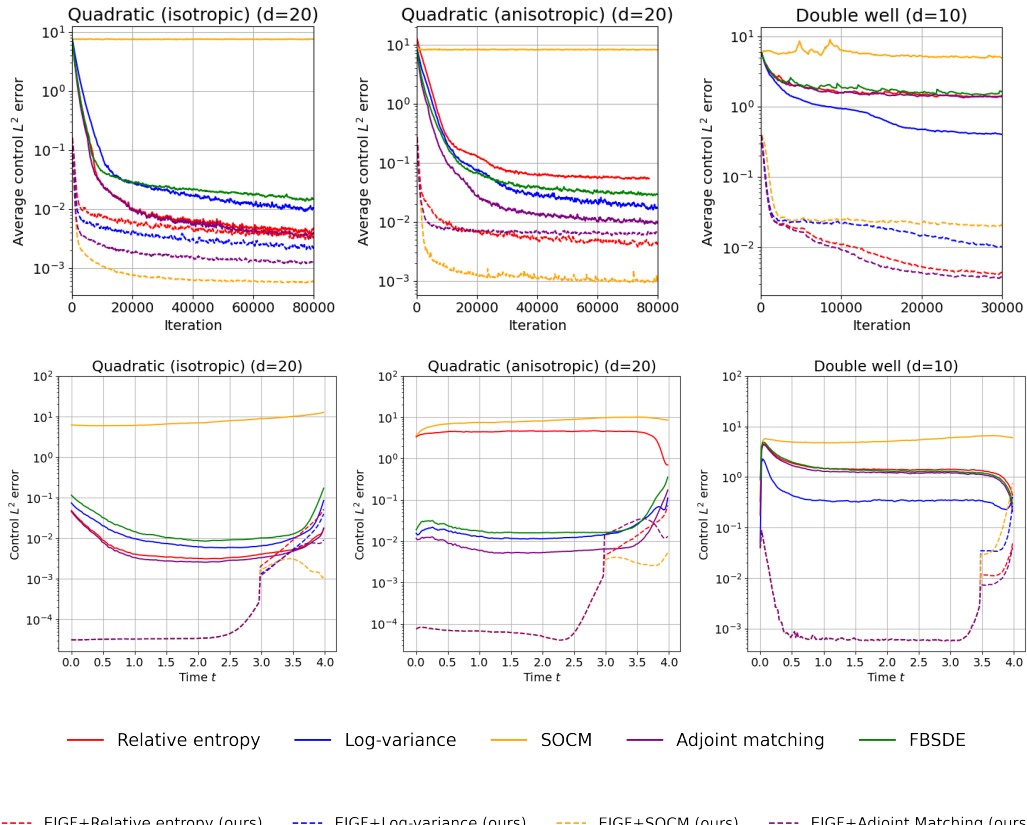

Figure 5: Average $L^2$ control error (EMA) as a function of iteration (top row) and $L^2$ error as a function of $t \in [0, T]$ (bottom row).

Figure 4 shows the results of the various eigenfunction losses. For the QUADRATIC settings, we can compute $\nabla \log \phi_0$ exactly, and see that the relative loss significantly improves upon existing loss functions for approximating this quantity (with the error measured in $L^2(\mu)$). For the other settings, the resulting control $\nabla \log \phi_0$ yields the lowest value of the control objective for the relative loss.

In Figure 5, we show the result of using the learned eigenfunctions in the IDO algorithm using the combined algorithm described in the previous section, and compare it with the standard IDO/FBSDE methods. In each setting, we obtain a lower $L^2$ error using the combined method, typically by an order of magnitude. The bottom row of Figure 5 shows how the error behaves as a function of $t \in [0, T]$: the pure eigenfunction method achieves superior performance for $t \to 0$, but performs worse closer to the terminal time $T$. The IDO method has constant performance in $[0, T]$, and the combined method combines the merits of both to provide the lowest overall $L^2$ error.

| Method | Algorithm/Loss | Control objective |
|---|---|---|
| EIGF (ours) | Relative loss | $\mathbf{73.33 \pm 0.02}$ |
| IDO | Log-variance loss | $74.52 \pm 0.02$ |
| IDO | Adjoint Matching | $74.69 \pm 0.02$ |
| IDO | Relative Entropy | $75.63 \pm 0.02$ |
| IDO | SOCM | Did not converge |

Table 1: Opinion dynamics, final control objective (smaller is better).

## 5.1 OPINION MODELING: DE GROOT MODEL

We additionally consider a networked control problem inspired by opinion formation. We model $N = 10$ agents with state $X_t \in \mathbb{R}^{10}$ and dynamics

$$dX_t = \big( (L - I)X_t - \gamma X_t + u(X_t) \big) \, dt + dW_t, \tag{23}$$

where $L \in \mathbb{R}^{10 \times 10}$ is a symmetric interaction matrix with $L_{ii} = 0.5$ and $L_{i,i\pm1} = 0.25$ (all other entries are zero), and we set $\gamma = 0.1$. We use the non-quadratic running cost

$$f(x) = \sum_{i=1}^{10} (x_i^2 - 1)^2, \tag{24}$$

and terminal cost $g(x) = 0$, with parameters $\lambda = 1.0$ and time horizon $T = 10.0$. Table 1 reports the final control objective (mean $\pm$ std) after 80,000 training iterations.

## 6 CONCLUSION

In this work, we have introduced a new perspective on a class of stochastic optimal control problems with gradient drift, showing that the optimal control can be obtained from the eigensystem of a Schrödinger operator. We have investigated the use of deep learning methods to learn the eigensystem, introducing a new loss function for this task. We have shown how this approach can be combined with existing IDO methods, yielding an improvement in $L^2$ error of roughly an order of magnitude over state-of-the-art methods in several long-horizon experiments, and overcoming the increase in computation cost typically associated with longer time horizons.

**Limitations** The main drawback of the proposed approach is that it is currently limited to problems with gradient drift. When the operator $\mathcal{L}$ is not even symmetric, it may no longer have real eigenvalues. Nonetheless, the top eigenfunction may still be real and nondegenerate with real eigenvalue, so that the long-term behaviour of the control is still described by an eigenfunction (Evans, 2022, Theorem 6.3). A second limitation is that there is no a priori method for determining an appropriate cutoff time $T_{cut}$, this is a hyperparameter that should be decided based on the application and the spectral gap $\lambda_1 - \lambda_0$.

## ACKNOWLEDGEMENTS

The work is supported by ETH research grant, Swiss National Science Foundation (SNSF) Project Funding No. 200021-207343, and SNSF Starting Grant. Artur Goldman is supported by the ETH AI Center through an ETH AI Center doctoral fellowship.

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

CONTENTS

## A  TECHNICAL DETAILS/ASSUMPTIONS

### A.1  REGULARITY CONDITIONS

Following Fleming & Soner (2006), we make the following assumptions, which guarantee that the SDE (3) has a unique strong solution.

1. The coefficient $b$ is Lipschitz in and satisfies the *linear growth* condition

$$\exists C > 0 : \|b(x) - b(y)\| \leq C\|x - y\|. \tag{25}$$

$$\exists C' > 0 : \|b(x)\| \leq C'(1 + \|x\|). \tag{26}$$

### A.2  SIMPLIFYING ASSUMPTION: $\sigma\sigma^T = I$

Since $\sigma \in \mathbb{R}^{d\times d}$ was assumed invertible, the matrix $\sigma\sigma^T$ is positive definite, and hence there exists a diagonalization $\sigma\sigma^T = U\Lambda U^T$ where $UU^T = I$ and $\Lambda = \text{diag}((\lambda_i)_{i=1}^d)$ for $\lambda_i > 0$. Consider now the change of variables $y = \Lambda^{-1/2}U^T x$. Then it holds that

$$\nabla_x \psi = U\Lambda^{-1/2}\,\nabla_x\psi, \quad \nabla_x^2\psi = U\Lambda^{-1/2}\,\nabla_y^2\psi\,\Lambda^{-1/2}U^T \tag{27}$$

and in particular $\text{Tr}(\sigma\sigma^T\nabla_x^2\psi) = \text{Tr}(U\Lambda U^T\nabla_x^2\psi) = \text{Tr}(\nabla_y^2\psi) = \Delta_y\psi$. Thus the PDE (8) can be written in terms of $y$ as

$$\partial_t\psi + \left(-\Delta_y - 2\beta\,b^T U\Lambda^{-1/2}\nabla_y + 2\beta^2 f\right)\psi = 0 \tag{28}$$

## B  PROOFS/DERIVATIONS

### B.1  COLE-HOPF TRANSFORMATION (8)

Let $V$ denote the value function defined in (5), satisfying the HJB equation (6). The so-called *Cole-Hopf transformation* consists of setting $V = -\beta^{-1}\log\tilde{\psi}$, leading to a linear PDE for $\tilde{\psi}$, which is called the *desirability function*. To obtain the form (8), we set $V(x,t) = -\beta^{-1}\log\psi\left(x, \frac{1}{2\beta}(T-t)\right)$. The derivatives of $V$ and $\psi$ are then related through

$$\partial_t V = \frac{1}{2\beta^2}\frac{\partial_t\psi}{\psi}, \quad \nabla V = -\beta^{-1}\frac{\nabla\psi}{\psi}, \tag{29}$$

$$\frac{\partial^2 V}{\partial x_i \partial x_j} = -\beta^{-1}\frac{1}{\psi^2}\left(\psi\frac{\partial^2\psi}{\partial x_i\partial x_j} - \left(\frac{\partial\psi}{\partial x_i}\frac{\partial\psi}{\partial x_j}\right)\right), \tag{30}$$

$$\implies \text{Tr}(\sigma\sigma^T\nabla^2 V) = \sum_{i,j}\sigma_{ij}^2\frac{\partial^2 V}{\partial x_i\partial x_j} = -\beta^{-1}\frac{1}{\psi}\text{Tr}(\sigma\sigma^T\nabla^2\psi) + \beta^{-1}\frac{1}{\psi^2}\|\sigma^T\nabla\psi\|^2. \tag{31}$$

Plugging these expressions into (6) gives

$$\frac{1}{\psi}\left(\frac{1}{2\beta^2}\partial_t\psi - \frac{1}{2\beta^2}\text{Tr}(\sigma\sigma^T\nabla^2\psi) - \beta^{-1}b^T\nabla\psi\right) + f = 0. \tag{32}$$

Multiplying with $2\beta^2\psi$ and combining with the terminal condition $V(x,T) = g(x)$ shows that the linear PDE for $\psi$ given in (8) is equivalent to the HJB equation (6).

### B.2  PROOF OF THEOREM 1

We begin with an informal argument. Recall that we are interested in solving the abstract evolution equation

$$\text{Find } \psi : [0,T] \to \mathcal{D}(\mathcal{L}) \text{ such that } \begin{cases} \partial_t\psi(t) + \mathcal{L}\psi(t) & = 0, \\ \psi(0) & = \psi_0. \end{cases} \tag{33}$$

Now, since $(\phi_i)_{i \in \mathbb{N}}$ forms an orthonormal basis, it holds that $\psi(t) = \sum_{i \in \mathbb{N}} a_i(t)\phi_i$, where $a_i(t) = \langle \psi_i(t), \phi \rangle$. Hence the PDE in (33) becomes

$$\sum_{i \in \mathbb{N}} \left( \frac{\mathrm{d}a_i(t)}{\mathrm{d}t} + \lambda_i a_i(t) \right) \phi_i = 0 \tag{34}$$

after formally interchanging the series expansion and derivatives. Since the $\phi_i$ are orthogonal, this equation is satisfied if and only if $a_i(t) = a_i(0)e^{-\lambda_i t}$ for each $i$.

Formally establishing (9) can be achieved through the theory of semigroups. Essentially, we want to formally define the semigroup $\left( e^{-t\mathcal{L}} \right)_{t \geq 0}$ and show that it forms the solution operator to (33). We begin by recalling the following definition.

**Definition 2** *The set of bounded linear operators on $\mathcal{H}$ is given by*

$$\mathcal{B}(\mathcal{H}) = \left\{ \mathcal{L} : \mathcal{H} \to \mathcal{H} : \mathcal{L} \text{ is linear and } \|\mathcal{L}\|_{op} = \sup_{f \in \mathcal{H}, \|f\|=1} \|\mathcal{L}(f)\| < \infty \right\} \tag{35}$$

The following result, sometimes called the *functional calculus form* of the spectral theorem, allows us to define $h(\mathcal{L})$ for bounded Borel functions $h$.

**Lemma 1** *(Reed & Simon, 1980, Theorem VIII.5)* Suppose $\mathcal{L}$ is a self-adjoint operator on $\mathcal{H}$. Then there exists a unique map $\hat{\phi}$ from the bounded Borel functions on $\mathbb{R}$ into $\mathcal{B}(\mathcal{H})$ which satisfies

- If $\mathcal{L}\psi = \lambda\psi$, then $\hat{\phi}(h)\psi = h(\lambda)\psi$.

In particular, when $\mathcal{L}$ admits a countable orthonormal basis of eigenfunctions $(\phi_i)$ with eigenvalues $\lambda_i$, this operator is given by

$$\hat{\phi}(h)\psi = h(\mathcal{L})\psi := \sum_{i \in \mathbb{N}} h(\lambda_i)\langle \phi_i, \psi \rangle \phi_i. \tag{36}$$

The next result makes use of the defined semigroup and leads to the desired representation. Note that we indeed consider a bounded function of the operator, as we consider it only for $t \geq 0$ and assume that the operator is bounded from below.

**Theorem 5** *(Reed & Simon, 1980, Theorem VIII.7)* Suppose $\mathcal{L}$ is self-adjoint and bounded from below, and define $T(t) = e^{-t\mathcal{L}}$ for $t \geq 0$. Then

(a) $T(0) = I$.

(b) For every $\phi \in D(\mathcal{L})$, it holds that

$$\left( \frac{\mathrm{d}}{\mathrm{d}t} T(t)\psi \right) \bigg|_{t=0} = \lim_{h \to 0} \frac{T(0+h)\psi - T(0)\psi}{h} = -\mathcal{L}T(0)\psi = -\mathcal{L}\psi \tag{37}$$

*Proof of Theorem 1* Combining (a) and (b) of the Theorem 5 shows that $\psi(t) := T(t)\psi_0$ satisfies

$$\frac{\mathrm{d}}{\mathrm{d}t}\psi(t) = \left( \frac{\mathrm{d}}{\mathrm{d}h} T(t+h)\psi_0 \right) \bigg|_{h=0} = -\mathcal{L}T(t)\psi_0 = -\mathcal{L}\psi(t), \quad \psi(0) = T(0)\psi_0 = \psi_0, \tag{38}$$

which is exactly the claim of Theorem 1. The last remark we make is that in Theorem 1 we only assume essential self-adjointness of $\mathcal{L}$, while above results operate with self-adjoint operators. Thus, the above theorems hold true for the closure $\overline{\mathcal{L}}$ of the operator $\mathcal{L}$. However, note that $\mathcal{L}$ is densely defined and has eigenfunctions which make up an orthonormal basis of both its domain and $L^2(\mu)$. Thus, the same representation (9) as for $\overline{\mathcal{L}}$ holds for $\mathcal{L}$. $\qquad\square$

### B.3 UNITARY EQUIVALENCE (11)

First, notice that $U : L^2(\mu) \to L^2(\mathbb{R}^d) : \psi \mapsto e^{-\beta E}\psi$ is indeed a unitary transformation, since

$$\forall \psi, \varphi \in L^2(\mu) : \langle U\psi, U\varphi \rangle_{L^2(\mathbb{R}^d)} = \int e^{-2\beta E} \psi\varphi \, \mathrm{d}x = \langle \psi, \varphi \rangle_{\mu}. \tag{39}$$

To establish equivalence, we compute

$$\nabla(e^{\beta E}\psi) = e^{\beta E} \left( \nabla \psi + \beta \nabla E \, \psi \right), \tag{40}$$

$$\Delta(e^{\beta E}\psi) = e^{\beta E}(\Delta \psi + 2\beta \langle \nabla E, \nabla \psi \rangle + \left( \beta \Delta E + \beta^2 \|\nabla E\|^2 \right) \psi), \tag{41}$$

$$2\beta \langle \nabla E, \nabla(e^{\beta E}\psi) \rangle = e^{\beta E}(2\beta \langle \nabla E, \nabla \psi \rangle + 2\beta^2 \|\nabla E\|^2 \psi). \tag{42}$$

Putting this together gives

$$\mathcal{L}(U^{-1}\psi) = \mathcal{L}(e^{\beta E}\psi) = e^{\beta E} \left( -\Delta \psi + \beta^2 \|\nabla E\|^2 \psi - \beta \Delta E \, \psi \right) + 2\beta^2 f e^{\beta E}\psi, \tag{43}$$

from which the result (11) follows.

### B.4 ESSENTIAL SELF-ADJOINTNESS OF $\mathcal{L}$

We will first show the following relation

$$\langle \varphi, \mathcal{L}\psi \rangle_{\mu} = \langle \nabla \varphi, \nabla \psi \rangle_{\mu} + 2\beta^2 \langle \varphi, f\psi \rangle_{\mu} \tag{44}$$

from which it is clear that $\mathcal{L}$ is symmetric on $C_0^\infty(\mathbb{R}^d)$. Indeed, using the divergence theorem, for $\psi, \phi \in C_0^\infty(\mathbb{R}^d)$ one obtains that

$$\langle \psi, -\Delta\varphi \rangle_{\mu} = -\int \psi\Delta\varphi \, e^{-2\beta E} \mathrm{d}x \tag{45}$$

$$= \int \left( \langle \nabla\psi, \nabla\varphi \rangle - 2\beta\psi \langle \nabla\varphi, \nabla E \rangle \right) e^{-2\beta E} \mathrm{d}x \tag{46}$$

$$= \langle \nabla\psi, \nabla\varphi \rangle_{\mu} - 2\beta \langle \psi, \langle \nabla E, \nabla\varphi \rangle \rangle_{\mu}, \tag{47}$$

which immediately shows the result.

While for matrices the notions of symmetry and self-adjointness are equivalent, the situation becomes more delicate for general (possibly unbounded) linear operators. In our case we can use the following result on the essential self-adjointness of the Schrödinger operator.

**Lemma 2** *(Reed & Simon, 1975, Theorem X.28) Let $\mathcal{V} \in L_{loc}^2(\mathbb{R}^d)$ with $\mathcal{V} \geq 0$ pointwise. Then $\mathcal{S} = -\Delta + \mathcal{V}$ is essentially self-adjoint on $C_0^\infty(\mathbb{R}^d)$.*

Note, that $C_0^\infty(\mathbb{R}^d)$ is dense in $L^2(\mathbb{R}^d)$. Thus, operator $\mathcal{L}$ is also densely defined as its domain contains $U^{-1}(C_0^\infty(\mathbb{R}^d))$.

The essential self-adjointness of $\mathcal{L}$ follows from the unitary equivalence with the Schrödinger operator as unitary transformation preserves essential self-adjointness of the transformed operator.

### B.5 PROOF OF THEOREM 2

As we have shown, the operator $\mathcal{L}$ is unitarily equivalent to the Schrödinger operator defined through

$$\mathcal{S} : D(\mathcal{S}) \subset L^2(\mathbb{R}^d) \to L^2(\mathbb{R}^d) : h \mapsto -\Delta h + \mathcal{V}h \tag{48}$$

with $\mathcal{V} = \beta^2 \|\nabla E\|^2 - \beta\Delta E + 2\beta^2 f$. All the properties mentioned in the statement of Theorem 2 remain unchanged under unitary equivalence (both the assumptions on the operator and the conclusions regarding the spectral properties). Hence it is sufficient to show the result for the Schrödinger operator $\mathcal{S}$ associated with $\mathcal{L}$.

This is precisely the content of the following theorem, proven in Reed & Simon (1978), combined with observations in section B.4.

**Theorem 6** *(Reed & Simon, 1978, Theorem XIII.67, XIII.64, XIII.47)* Suppose $\mathcal{V} \in L^1_{loc}(\mathbb{R}^d)$ is bounded from below and $\lim_{|x| \to \infty} \mathcal{V}(x) = \infty$. Then the Schrodinger operator $\mathcal{S} = -\Delta + \mathcal{V}$ has compact resolvent, and in particular a purely discrete spectrum and an orthonormal basis of eigenfunctions in $L^2(\mathbb{R}^d)$. The spectrum $\sigma(\mathcal{S})$ is bounded from below and the eigenvalues do not have a finite accumulation point. If in addition $\mathcal{V} \in L^2_{loc}(\mathbb{R}^d)$, the smallest eigenvalue has multiplicity one, and the corresponding eigenfunction can be taken to be strictly positive.

### B.6 PROOF OF THEOREM 4

The result relies on the fact that the Schrödinger operator associated with $\mathcal{L}$ has a quadratic potential, for which the eigenfunctions can be computed exactly. Indeed, the following is a standard result that can be found in many textbooks on quantum mechanics (Griffiths & Schroeter, 2018):

**Lemma 3** *Consider the Schrödinger operator in $d = 1$ with quadratic potential,*

$$\mathcal{S} : D(\mathcal{S}) \subset L^2(\mathbb{R}) \to L^2(\mathbb{R}) : \psi \mapsto -\frac{\mathrm{d}^2\psi}{\mathrm{d}x^2} + x^2\psi. \tag{49}$$

*The eigenvalues of $\mathcal{S}$ are given by $\lambda_n = 2n + 1$, and the normalized eigenfunctions are given by*

$$\phi_n(x) = \frac{1}{\pi^{1/4}} \frac{1}{\sqrt{2^n n!}} H_n(x) e^{-x^2/2} \tag{50}$$

*where $H_n$ denotes the $n$-th* physicist's Hermite polynomial *defined through*

$$H_0(x) = 1, \quad H_1(x) = 2x, \quad H_{n+1}(x) = 2xH_n(x) - 2nH_{n-1}(x). \tag{51}$$

*These satisfy the generating function relation*

$$e^{2xt-t^2} = \sum_{n=0}^{\infty} H_n(x) \frac{t^n}{n!}. \tag{52}$$

This result can be easily extended to higher dimensions:

**Lemma 4** *Consider now the $d$-dimensional Harmonic oscillator:*

$$\mathcal{S} : D(\mathcal{S}) \subset L^2(\mathbb{R}^d) \to L^2(\mathbb{R}^d) : \psi \mapsto -\Delta\psi + x^T A x\, \psi. \tag{53}$$

*where $A \in \mathbb{R}^{d \times d}$ is symmetric and positive definite. Denote $A = U^T \Lambda U$ the diagonalization of A. Then the eigenvalues are indexed by the multi-index $\alpha \in \mathbb{N}^d$, and are given by*

$$\lambda_\alpha = \sum_{i=1}^{d} \Lambda_i^{1/2}(2\alpha_i + 1). \tag{54}$$

*The associated normalized eigenfunctions are*

$$\phi_\alpha(x) = \frac{1}{\pi^{d/4}} \exp\left(-\frac{1}{2}x^T U^T \Lambda^{1/2} U x\right) \prod_{i=1}^{d} \frac{\Lambda_i^{1/8}}{\sqrt{2_i^\alpha (\alpha_i)!}} H_{\alpha_i}((\Lambda^{1/4}Ux)_i) \tag{55}$$

*Proof.* We begin by introducing the variable $y = \Lambda^{1/4}Ux$. This gives

$$x^T A x = y^T \Lambda^{1/2} y \tag{56}$$

$$\Delta_x \psi = \mathsf{Tr}(\Lambda^{1/2} \nabla_y^2 \psi) \tag{57}$$

so that we can write

$$-\Delta_x \psi + x^T A x \psi = \sum_{i=1}^{d} \Lambda_i^{1/2} \left(-\frac{\partial^2\psi}{\partial y^2} + y_i^2\psi\right) \tag{58}$$

Thus, the $d$-dimensional harmonic oscillator decouples into $d$ rescaled one-dimensional oscillators. Since $L^2(\mathbb{R}^d) \cong \bigotimes_{i=1}^{d} L^2(\mathbb{R})$ (after completion), this means that the eigenfunctions are given by the

product of the one-dimensional eigenfunctions. Hence the spectrum is indexed by $\alpha \in \mathbb{N}^d$ and the eigenfunctions are, expressed in the variable $y$,

$$\phi_\alpha(y) = \frac{1}{\pi^{d/4}} \prod_{i=1}^d \frac{1}{\sqrt{2^{\alpha_i}(\alpha_i!)}} H_{\alpha_i}(y_i) e^{-y_i^2/2} \tag{59}$$

To get the eigenfunctions in the $x$ variable, we transform back and note that the above expression is normalized as $\int \phi_\alpha(y)^2 \mathrm{d}y = 1$, while we need $\int \phi_\alpha(x)^2 \mathrm{d}x = 1$. Using the fact that $\mathrm{d}y = \det(\Lambda^{1/4})\mathrm{d}x$, this yields the final normalized eigenfunctions as

$$\phi_\alpha(x) = \frac{1}{\pi^{d/4}} \exp\left(-\frac{1}{2}x^T U^T \Lambda^{1/2} U x\right) \prod_{i=1}^d \frac{\Lambda_i^{1/8}}{\sqrt{2_i^\alpha(\alpha_i)!}} H_{\alpha_i}((\Lambda^{1/4}Ux)_i) \tag{60}$$

and the eigenvalues as

$$\lambda_\alpha = \sum_{i=1}^d \Lambda_i^{1/2}(2\alpha_i + 1). \tag{61}$$

$\square$

Now consider the operator $\mathcal{L}$ as defined in (10), with $b(x) = -Ax$ and $f(x) = x^T P x$. Since $A$ is symmetric, it holds that $b = -\nabla E$ with $E(x) = \frac{1}{2}x^T A x$. It follows that $\mathcal{L}$ is unitarily equivalent with the Schrödinger operator with potential

$$\mathcal{V} = -\beta\Delta E + \beta^2\|\nabla E\|^2 + 2\beta^2 f = -\beta\mathsf{Tr}(A) + \beta^2 x^T(A^T A + 2P)x. \tag{62}$$

The first term gives a constant shift to the eigenvalues, but otherwise we are precisely dealing with the $d$-dimensional harmonic oscillator, whose eigensystem is described in Lemma 4. Hence the eigenvalues are precisely (14), and the eigenfunctions of the original operator $\mathcal{L}$ are then obtained by multiplying with $e^{\frac{\beta}{2}x^T A x}$, which yields (13).

### B.7 SEMIGROUP AND VISCOSITY SOLUTIONS

**Plan of the section.** We work with two linked equations on $\mathbb{R}^d \times [0, T]$: the HJB equation (6) and the linear parabolic PDE (8) obtained from (6) via the (monotone) Cole–Hopf transform. Our goal is to produce the optimal control $u^*$ by first solving the linear PDE, then applying the inverse Cole–Hopf transform to obtain a value function candidate $V'$. This program raises two issues which have to be addressed:

1. *Nonuniqueness in unbounded domains.* On $\mathbb{R}^d$ (even with the same terminal/boundary data), second-order finite-horizon HJB and linear parabolic equations may admit multiple solutions unless one restricts to an appropriate growth class; see Tychonoff (1935) for a classical example of this phenomenon.

2. *Verification.* To identify $u^*$ from a solution of (6) one uses a verification theorem. These theorems require the candidate $V'$ to be a (sufficiently regular) viscosity solution (Crandall et al., 1992) lying in a class where comparison (hence uniqueness) holds, see (Fleming & Soner, 2006, Section V.9) for an example of such result.

In our approach we fix a specific *semigroup solution* of the linear PDE and use it to define $V'$ via the inverse Cole–Hopf map. Specifically, we have $V'(x, t) = -\beta^{-1}\log\psi(x, \frac{1}{2\beta}(T - t))$, where $\psi(x, t) = (e^{-t\mathcal{L}}\psi_0)(x)$. The central task is therefore to (a) place this $V'$ in a uniqueness class for (6) and (b) confirm the hypotheses of a verification theorem. In the following text we establish sufficient conditions under which the task is solved.

**Verification theorem and Comparison (uniqueness) in $\mathbb{R}^d$ under growth constraints.** There are several ways to establish that the value function $V$ (5) is a unique solution of HJB equation (6) in some growth class for the given terminal conditions. In general it is obtained with a use of Dynamic Programming Principle. However, this approach is not so simple in the case of unbounded controls and non-smooth viscosity solutions. We refer to (Zhou et al., 1997; Fleming & Soner, 2006; Da Lio & Ley, 2006; 2011) for an overview of ways to establish connection between value function and

viscosity solutions, as well as results regarding existence and uniqueness of viscosity solutions in appropriate growth classes. Following Da Lio & Ley (2011) we introduce class of functions $\tilde{\mathcal{C}}_p$. We say that a locally bounded function $u : \mathbb{R}^d \times [0; T] \to \mathbb{R}$ is in the class $\tilde{\mathcal{C}}_p$ if for some $C > 0$ we have

$$|u(x,t)| \leq C(1 + \|x\|^p), \; \forall \, (x,t) \in \mathbb{R}^d \times [0; T]. \tag{63}$$

Now, we can formulate the following

**Theorem 7** *(Da Lio & Ley, 2006, Thm. 3.1)+(Da Lio & Ley, 2011, Thm. 3.2)* Suppose that in (6) we have

- $b$ satisfies (25) and (26), and $\sigma = I$
- Running cost $f$ satisfies: $\exists C_1 > 0 : \forall x \in \mathbb{R}^d : |f(x)| \leq C_1(1 + \|x\|^p)$
- Terminal condition satisfies $g \in \tilde{\mathcal{C}}_p$

Then there exists $0 \leq \tau < T$ such that (6) has a unique continuous viscosity solution in $\mathbb{R}^d \times [\tau; T]$ in the class $\tilde{\mathcal{C}}_p$. Moreover, this unique solution is the value function $V$ (5).

**Remark 3** *See discussion in (Da Lio & Ley, 2011, Remark 3.1) why we can't hope to obtain existence in Theorem 7 on a whole interval $[0; T]$.*

**Growth of semigroup solution** From previous paragraphs it is clear that one way to provide a link between proposed $V'$ and solutions of (6) is to to establish growth rates on $V'$. Specifically, following Theorem 7 we want to show that $V' \in \tilde{\mathcal{C}}_p$.

There are other ways to establish connection between solutions, see Biton (2001), (Fleming & Soner, 2006, Section VI). However, we choose to pursue a path related to growth conditions of spectral elements of Schrödinger operator (Davies & Simon, 1984; Baraniewicz, 2024).

**Theorem 8** *(Davies & Simon, 1984, Thm. 6.1, Thm. 6.3)* Let $\mathcal{S} = -\Delta + \mathcal{V}$ on $\mathbb{R}^d$ with ground state $\phi_0 > 0$ (normalized in $L^2(\mathbb{R}^d)$) and suppose that there exist $C_1, C_3 > 0$ and $C_2, C_4 \in \mathbb{R}$ such that, for all $x$ with $\|x\|$ large,

$$C_3\|x\|^b + C_4 \; \leq \; \mathcal{V}(x) \; \leq \; C_1\|x\|^a + C_2, \qquad \text{where} \quad \tfrac{a}{2} + 1 < b \leq a.$$

Then $\mathcal{S}$ is *intrinsically ultracontractive (IUC)*. Moreover, there exist constants $C_5, C_7 > 0$ and $C_6, C_8 \in \mathbb{R}$ such that, as $\|x\| \to \infty$,

$$C_5 \|x\|^{\frac{b}{2}+1} + C_6 \; \leq \; -\log \phi_0(x) \; \leq \; C_7 \|x\|^{\frac{a}{2}+1} + C_8. \tag{64}$$

*Proof.* Intrinsic ultracontractivity under the stated growth is exactly (Davies & Simon, 1984, Thm. 6.3). The upper growth bound in (64) is the estimate $-\log \phi_0(x) \leq C\|x\|^{\frac{a}{2}+1}$ stated explicitly in (Davies & Simon, 1984, Eq. (6.4)). For the lower bound in (64), take a comparator $W(x) = \tilde{c} \|x\|^b$ with $0 < \tilde{c} < C_3$; then $W \to \infty$ and $\mathcal{V} - W \to \infty$, so the ground states satisfy $\phi_0^{(\mathcal{V})} \leq C \phi_0^{(W)}$ by the comparison Lemma (Davies & Simon, 1984, Lem. 6.2). (Davies & Simon, 1984, App. B) constructs WKB-type barriers for $-\Delta + W$ and gives pointwise upper bounds of the form $\phi_0^{(W)}(x) \leq C'\|x\|^{-\beta} \exp\{-\kappa\|x\|^{1+\frac{b}{2}}\}$ for large $\|x\|$ (see the JWKB ansatz and subharmonic comparison in (Davies & Simon, 1984, App. B, esp. Lem. B.1–B.3)); combining yields the stated lower bound on $-\log \phi_0$. $\qquad \square$

**Remark 4** *Note that we use ultracontractivity properties of $\mathcal{S}$ which require growth of $\mathcal{V}$ to be at least as $\|x\|^{2+\varepsilon}, \varepsilon > 0$. The characterization of operator contractivity properties is fully given in (Davies & Simon, 1984, Thm. 6.1) and does not allow for slower orders of growth. This restriction on growth of $\mathcal{V}$ is encoded through condition $\frac{a}{2} + 1 < b \leq a$.*

**Theorem 9** *Let*

$$\mathcal{V} = \beta\|\nabla E\|^2 - \Delta E + 2\beta f, \qquad \mathcal{S} = U\mathcal{L}U^{-1} = -\Delta + \mathcal{V}.$$

where $U^{-1} f = e^{\beta E} f$. Let $\psi(x, t) = (e^{-t\mathcal{L}} \psi_0)(x)$ and define the candidate value function

$$V'(x, t) = -\beta^{-1} \log \psi \left( x, \tfrac{1}{2\beta}(T - t) \right).$$

Assume the hypotheses of Theorem 8 for $\mathcal{V}$. Finally, let us suppose that there exist $0 < m \leq M$ so that

$$m \leq \frac{\psi_0}{U^{-1} \phi_0} \leq M, \ \forall x \in \mathbb{R}^d, \tag{65}$$

where $\phi_0$ is a ground state of Shrödinger operator $\mathcal{S}$. Then, uniformly for all $t \in [0, T]$, there exist constants $C_1, C_3 > 0$ and $C_2, C_4 \in \mathbb{R}$ such that, as $\|x\| \to \infty$,

$$C_1 \|x\|^{\frac{b}{2}+1} + C_2 \ \leq \ V'(x, t) + E(x) \ \leq \ C_3 \|x\|^{\frac{a}{2}+1} + C_4. \tag{66}$$

*Proof.* Recall that the ground state of $\mathcal{L}$ is an eigenfunction $\varphi_0 = U^{-1} \phi_0$, where $\phi_0$ is a ground state of Shrödinger operator $\mathcal{S}$. Let us introduce the ground state transformed semigroup:

$$P_t^{\varphi_0} g(x) := \frac{e^{\lambda_0 t}}{\varphi_0(x)} (e^{-t\mathcal{L}} (\varphi_0 g))(x)$$

Thus, we have a representation

$$\psi(x, t) = (e^{-t\mathcal{L}} \psi_0)(x) = e^{-\lambda_0 t} \varphi_0(x) P_t^{\varphi_0} \left( \frac{\psi_0}{\varphi_0} \right)(x)$$

It is known, that semigroup $P_t^{\varphi_0} g(x)$ is Markov, see Kaleta et al. (2018). Using the fact that $P_t^{\varphi_0} g(x)$ is Markov and (65) we have the two-sided bound for all $t \in [0; T]$

$$e^{-\lambda_0 t} \varphi_0(x) m \leq (e^{-t\mathcal{L}} \psi_0)(x) \leq e^{-\lambda_0 t} \varphi_0(x) M$$

which leads to a bound

$$\lambda_0 \frac{T - t}{2\beta^2} - \beta^{-1} \log M - \beta^{-1} \log \phi_0 - E(x) \leq V'(x, t)$$

$$\leq \lambda_0 \frac{T - t}{2\beta^2} - \beta^{-1} \log m - \beta^{-1} \log \phi_0 - E(x) \quad (67)$$

Applying growth bounds on ground state of Shrödinger operator from Theorem 8 leads to the desired bound. $\qquad \square$

**Remark 5** *Note that definition of $\tilde{\mathcal{C}}_p$ (63) requires a growth bound on coordinate $x$ uniformly for all $t \in [0; T]$. Unfortunately, analysis based purely on spectral properties of operator leads to non-uniform growth bounds depending on $t$, which blow up when $t \to 0$ for $e^{-t\mathcal{L}}$ (for example, note that constants in (Davies & Simon, 1984, Thm. 3.2) depend on time). For this reason and for simplicity we introduced assumption (65) on boundary conditions.*

**Relationship between semigroup-based and viscosity solution**    Finalising everything in this section, we formulate the following result which addresses the questions raised above.

**Theorem 10** *Consider HJB equation (6). Let $\mathcal{V} = \beta \|\nabla E\|^2 - \Delta E + 2\beta f$. Assume the following:*

- *$b(x) = -\nabla E(x)$ satisfies (25) and (26), and $\sigma = I$.*

- *$\mathcal{V}$ satisfies Assumption **(A2)**. Moreover, there exist $C_1, C_3 > 0$ and $C_2, C_4 \in \mathbb{R}$ such that, for all $x$ with $\|x\|$ large,*

$$C_3 \|x\|^b + C_4 \ \leq \ \mathcal{V}(x) \ \leq \ C_1 \|x\|^a + C_2, \qquad \text{where} \quad \tfrac{a}{2} + 1 < b \leq a. \tag{68}$$

- *Let $\phi_0$ be the ground state of operator $\mathcal{S} = -\Delta + \mathcal{V}$. Assume $g \in \tilde{\mathcal{C}}_a$ and there exist $0 < m \leq M$ so that*

$$m \leq \frac{\exp(-\beta g)}{\exp(\beta E(x)) \phi_0} \leq M, \ \forall x \in \mathbb{R}^d, \tag{69}$$

*Then there exists $0 \leq \tau < T$ such that function $V'(x,t) = -\beta^{-1} \log \psi(x, \frac{1}{2\beta}(T-t))$, where $\psi(x,t) = (e^{-t\mathcal{L}}\psi_0)(x)$ and $\mathcal{L}$ and $\psi_0$ are from (8), is a unique continuous viscosity solution of (6) on $\mathbb{R}^d \times [\tau; T]$ in the class $\tilde{\mathcal{C}}_{a/2+1}$. Moreover, it coincides with value function $V$ (5) on $\mathbb{R}^d \times [\tau; T]$.*

*Proof.* Under Assumption **(A2)** Theorem 6 holds. Theorem 6 ensures that operators $\mathcal{S}$ and $\mathcal{L}$ have some good properties and ground state $\phi_0$ is positive.

Under the stated assumptions Theorem 7 and Theorem 9 hold. Under growth assumptions (26) and (68) Theorem 9 gives us that $V' \in \tilde{\mathcal{C}}_{a/2+1} \subseteq \tilde{\mathcal{C}}_a$ which has to be the unique viscosity solution in $\tilde{\mathcal{C}}_a$ given by Theorem 7 on $\mathbb{R}^d \times [\tau; T]$, which coincides with value function $V$ (5) on $\mathbb{R}^d \times [\tau; T]$. $\quad\square$

**Remark 6** *Note that assumption (69) is realistic under rapid growth of $\mathcal{V}$, taking in account growth bounds on the ground state (64) and $b(x) = \nabla E(x)$ in (26), and does not contradict $g \in \tilde{\mathcal{C}}_a$.*

## C  LOSS FUNCTIONS

### C.1  EXISTING METHODS FOR SHORT-HORIZON SOC

**Grid-based solvers**   In low dimensions ($d \leq 3$), classical techniques for numerically solving PDEs can be used. These include finite difference (Bonnans et al., 2004) and finite element methods (Jensen & Smears, 2013; Ern & Guermond, 2004; Brenner & Scott, 2008), as well as semi-Langrangian schemes (Calzola et al., 2023; Carlini et al., 2020) and multi-level Picard iteration (E et al., 2021).

**FBSDE solvers**   In another line of work, the SOC problem is transformed into a pair of forward-backward SDEs (FBSDEs). These are solved through dynamic programming (Gobet et al., 2005; Longstaff & Schwartz, 2001) or deep learning methods which parametrize the solution to the FBSDE using a neural network (Han et al., 2018; E et al., 2017; Andersson et al., 2023).

**IDO methods**   In recent years, many methods have been proposed which parametrize the control $u_\theta$ directly, and optimize it by rolling out simulations of the system (3) under the current control. Authors of Nüsken & Richter (2021) coined the term *iterative diffusion optimization (IDO)*, arguing that many of these methods can be viewed from a common perspective given in Algorithm 2 (Appendix E). This class of algorithms contains state-of-the art methods such as SOCM and adjoint matching (Holdijk et al., 2023; Domingo-Enrich et al., 2024a). We describe the most commonly used loss functions in Appendix C.

### C.2  EXTENDING TO MULTIPLE EIGENFUNCTIONS

**PINN loss**   The most common way to extend the PINN loss (15) to multiple eigenfunctions is to define

$$\mathcal{R}_{\text{PINN}}^k(\phi) = \sum_{i=0}^{k} \left( \|\mathcal{L}[\phi_i] - \lambda_i \phi_i\|_\rho^2 + \alpha_{norm}(\|\phi_i\|_\rho^2 - 1)^2 \right) + \alpha_{orth} \sum_{j \neq i} \langle \phi_i, \phi_j \rangle_\mu^2 \qquad (70)$$

for $\alpha_{norm}, \alpha_{orth} > 0$. Here we have denoted $\phi : \mathbb{R}^d \to \mathbb{R}^{k+1}$. The main difference is the addition of the orthogonal regularization term, which both ensures that the same eigenfunction is not learned twice, and attempts to speed up learning by enforcing the known property that eigenfunctions with different eigenvalues are orthogonal w.r.t. the inner product $\langle \cdot, \cdot \rangle_\mu$.

**Variational loss**   A similar idea is used to generalize the variational loss (17). The following result shows that the variational principle (16) can be extended to multiple eigenfunctions by imposing orthogonality.

**Theorem 11** *(Zhang et al., 2022, Theorem 1)* Let $k \in \mathbb{N}$, and let $\mathcal{L}$ be a self-adjoint operator with discrete spectrum which admits an orthonormal basis of eigenfunctions, and whose eigenvalues are bounded below. Furthermore, let $\omega_0 \geq \ldots \geq \omega_k > 0$ be real numbers. Then it holds that

$$\sum_{i=0}^{k} \omega_i \lambda_i = \inf_{f_0, \ldots, f_k \in \mathcal{H}} \sum_{i=0}^{k} \omega_i \langle f_i, \mathcal{L} f_i \rangle \qquad (71)$$

where the infimum is taken over all $(f_i)_{i=0}^k \subset \mathcal{H}$ such that

$$\forall i, j \in \{0, \dots, k\} : \langle f_i, f_j \rangle = \delta_{ij} \tag{72}$$

*Proof.* The proof of this result is given in Zhang et al. (2022). $\qquad\square$

Based on this result, the following generalization of the variational loss is proposed in Zhang et al. (2022):

$$\mathcal{R}_{\mathrm{Var}}^k(\phi) = \sum_{i=0}^k \langle \phi_i, \mathcal{L}\phi_i \rangle + \alpha \left\| \mathbb{E}_\mu \left[ \phi\phi^T \right] - I \right\|_F^2, \tag{73}$$

where we have written $\| \cdot \|_F$ for the Frobenius norm and $\alpha > 0$. This loss was also studied in Cabannes et al. (2023), where it was noted that the minimizers of the variational loss (17)-(73) are not obtained at the eigenfunctions. Instead, the following characterization of the minimizers of (73) was obtained.

**Lemma 5** *(Cabannes et al., 2023, Lemma 2)* Suppose that $\mathcal{H} = L^2(\mu)$. Then it holds that

$$\underset{\psi_0, \dots, \psi_k \in \mathcal{H}}{\arg\min} \ \mathcal{R}_{\mathrm{Var}}^k(\psi) = \left\{ U\tilde{\phi} \mid UU^T = I, \tilde{\phi}_i = \sqrt{\left(1 - \frac{\lambda_i}{2\alpha}\right)_+} \phi_i \right\} \tag{74}$$

where $(\phi_i, \lambda_i)$ denotes the orthonormal eigensystem of $\mathcal{L}$.

## C.3 Estimating $\lambda_i$

The main advantage of the variational loss (17), (73) is that it does not require the eigenvalues of $\mathcal{L}$ to be available. However, Lemma 5 shows that the minimizers of $\mathcal{R}_{\mathrm{Var}}^k$ do not coincide exactly with the eigenfunctions of $\mathcal{L}$, so we cannot naively compute $\langle \phi, \mathcal{L}\phi \rangle_\mu$ to obtain the eigenvalues. Instead, the following lemma shows how to obtain the eigenvalues and eigenfunctions from an element in the minimizing set (74).

**Lemma 6** *Suppose it holds that $\psi = U\tilde{\phi}$, where $UU^T = I$ and $\tilde{\phi}_i = \sqrt{\left(1 - \frac{\lambda_i}{2\alpha}\right)} \phi_i$ for each $i$, and $\alpha > \frac{\lambda_k}{2}$. Then the first $k+1$ eigenfunctions and eigenvalues are given by*

$$\phi = D^{-1/2} U^T \psi, \qquad \lambda_i = \frac{2}{\beta}(1 - D_{ii}) \tag{75}$$

*where $UDU^T = \mathbb{E}_\mu[\psi\psi^T]$ is the diagonalization of the second moment matrix of $\psi$.*

*Proof.* By definition of $\psi$, we have

$$\mathbb{E}_\mu[\psi\psi^T] = U\mathbb{E}_\mu[\tilde{\phi}\tilde{\phi}^T]U^T \tag{76}$$

$$= U\left(I - \frac{1}{2\alpha}\Lambda\right)U^T \tag{77}$$

where $\Lambda = \mathrm{diag}(\lambda_i)$ and we have used the orthonormality property of the eigenfunctions. From this, we obtain that

$$D = I - \frac{\beta}{2}\Lambda, \quad \psi = UD^{1/2}\phi \tag{78}$$

which concludes the proof. $\qquad\square$

In light of this result, we can estimate the second moment matrix $\mathbb{E}_\mu[\psi\psi^T]$, apply a diagonalization algorithm, and obtain the eigenfunctions and eigenvalues using (75).

## C.4 Empirical loss & sampling

**Rewriting variational loss** Recalling the equation (44),

$$\langle \varphi, \mathcal{L}\psi \rangle_\mu = \langle \nabla\varphi, \nabla\psi \rangle_\mu + 2\beta^2 \langle \varphi, f\psi \rangle_\mu, \tag{79}$$

we see that it is possible to evaluate inner products of the form $\langle \varphi, \mathcal{L}\psi \rangle_\mu$ by only evaluating $\varphi, \psi$ and its derivatives. This avoids the expensive computation of the second order derivatives of the neural network, making the variational losses less memory-intensive than the other loss functions, which requires explicit computation of $\mathcal{L}\phi$.

**Estimation of inner products** All of the loss functions discussed for learning eigenfunctions contain inner products of the form $\langle \psi, \phi \rangle_\mu$. To obtain an empirical loss, these quantities most be approximated. When $\mu$ is a density, this is done using a Monte Carlo estimate

$$\langle \varphi, \psi \rangle_\mu = \int \varphi \psi \, d\mu = \mathbb{E}_\mu[\varphi \psi] \approx \frac{1}{m} \sum_{i=1}^m \varphi(X_i) \psi(X_i), \quad X_i \sim \mu. \tag{80}$$

Since $\mu(x) \propto \exp(-2\beta E(x))$, we can employ Markov Chain Monte Carlo (MCMC) techniques in order to obtain the samples $(X_i)$ (Brooks et al., 2011). In particular, when training the eigenfunctions, we can store $m$ samples in memory and apply some number of MCMC steps in each iteration to update these samples. Alternatively, one can pre-sample a large dataset of samples from $\mu$.

**Non-confining energy.** In general, $\mu$ may not be a finite measure, for instance when we have a repulsive LQR ($E = -\frac{1}{2}\|x\|^2$). In this case, the inner products can no longer be estimated directly from samples, but may still be computed by using importance sampling techniques (Liu, 2004; Tokdar & Kass, 2010). In the case where the measure defined through $\mu(x) = \exp(-2\beta E(x))$ is not finite but $\bar{\mu}(x) = \exp(2\beta E(x))$ is, we can apply importance sampling with $\bar{\mu}$, so that the inner products are obtained as

$$\langle \varphi, \psi \rangle_\mu = \int \varphi \psi e^{-2\beta E(x)} dx = \mathbb{E}_{\bar{\mu}}\left[\bar{\varphi}\bar{\psi}\right] \tag{81}$$

where we have defined $\bar{\varphi} = e^{-2\beta E}\varphi$ and $\bar{\psi} = e^{-2\beta E}\psi$. For stable training, it is then advisable to parametrize $\bar{\phi} := \phi e^{-2\beta E}$ instead of $\phi$.

## C.5 Logarithmic regularization

As discussed in the main text, the PINN and relative eigenfunction losses (15)-(20) have the form

$$\mathcal{R}(\phi) = \mathcal{R}_{main}(\phi) + \alpha \mathcal{R}_{reg}(\phi), \quad \mathcal{R}_{reg}(\phi) = (\|\phi\|_\rho^2 - 1)^2. \tag{82}$$

for some $\alpha > 0$. We observe in our experiments that this regularizer is sometimes not strong enough, and the network may still converge to $\phi = 0$. For this reason, we instead use a logarithmic regularizer

$$\mathcal{R}_{reg}(\phi) = (\log \|\phi\|_\rho)^2 \tag{83}$$

The behaviour is exactly the same as before for $\|\phi\|_\rho \approx 1$, since $\log(1+x) = x + \mathcal{O}(x^2)$ as $x \to 0$, but this regularizer avoids the convergence to $\phi = 0$.

## C.6 FBSDE Loss

We briefly discuss the Robust FBSDE loss for stochastic optimal control introduced in Andersson et al. (2023). The main idea is that the solution to the HJB equation (6) can be written down as a pair of SDEs, as the following lemma illustrates.

**Lemma 7** *Suppose $\sigma = I$. Then it holds that the solution to the pair of FBSDEs*

$$dX_t = (b(X_t, t) - Z_t)dt + \sqrt{\lambda}dW_t, \qquad\qquad X_0 \sim p_0, \tag{84}$$

$$dY_t = \left(-f(X_t, t) - \frac{1}{2}\|Z_t\|^2\right)dt + \sqrt{\lambda}\langle Z_t, dW_t \rangle, \qquad Y_T = g(X_T) \tag{85}$$

*is given by $Z_t = \partial_x V(X_t, t)$ and $Y_t = V(X_t, t)$.*

This pair of SDEs can be transformed in a variational formulation which is amenable to deep learning, as described in detail in Andersson et al. (2023). Using the Markov property of the FBSDE, one can show that $Z_t = \zeta(t, X_t)$ for some function $\zeta : [0, T] \times \mathbb{R}^d \to \mathbb{R}^d$, and that the FBSDE problem can be reformulated as the following variational problem:

$$\begin{cases} \text{minimize}_\zeta \ \Psi_\alpha(\zeta) = \mathbb{E}[\mathcal{Y}_0^\zeta] + \alpha \mathbb{E}\left[|Y_T^\zeta - g(X_T^\zeta)|\right], \quad \text{where} \\ \mathcal{Y}_0^\zeta = g(X_T^\zeta) + \int_0^T \left(f(X_t^\zeta, t) + \frac{1}{2}\|Z_t^\zeta\|^2\right)dt - \int_0^T \langle Z_t^\zeta, dW_t \rangle \\ X_t^\zeta = X_0 + \int_0^t \left(b(s, X_s^\zeta) - Z_s^\zeta\right)dt + \int_0^t \lambda dW_s \\ Y_t^\zeta = \mathbb{E}[\mathcal{Y}_0^\zeta] - \int_0^t \left(f(X_s^\zeta, s) + \frac{1}{2}\|Z_s^\zeta\|^2\right)ds + \int_0^t \langle Z_s^\zeta, dW_s \rangle, \quad Z_t^\zeta = \zeta(t, X_t^\zeta), \quad t \in [0, T] \end{cases} \tag{86}$$

Parametrizing $\zeta$ as a neural network, we can then simulate the stochastic processes in (86) and define the loss function as

$$\mathcal{R}_{FBSDE}(u; X^u) = \mathbb{E}[\mathcal{Y}_0^\zeta] + \alpha \mathbb{E}\left[|Y_T^\zeta - g(X_T^\zeta)|\right], \quad \text{with } \zeta = -u \tag{87}$$

For more details we refer to the relevant work Andersson et al. (2023).

### C.7 IDO LOSS FUNCTIONS

The IDO algorithm described in Algorithm 2 (Appendix D) is rather general, and a large number of algorithms can be obtained by specifying different loss functions. For a detailed discussion of the various loss functions and the relations between the resulting algorithms, we refer to Domingo-Enrich (2024); Domingo-Enrich et al. (2024b). Here, we go over the loss functions that were used for the experiments in this paper.

**Adjoint loss** The most straightforward choice of loss is to simply use the objective of the control, and define

$$\mathcal{R}(u; X^u) = \int_0^T \left(\frac{1}{2}\|u(X_t^u, t)\|^2 + f(X_t^u)\right) dt + g(X_T^u). \tag{88}$$

This loss is also called the *relative entropy loss*. When converting this to an empirical loss, there are two options. The *discrete adjoint method* consists of first discretizing the objective and then differentiating w.r.t. the parameters. However, this requires storing the numerical solver in memory and can hence be quite memory-intensive.

The *continuous adjoint method* instead analytically computes derivatives w.r.t. the state space. To this end, define the adjoint state as

$$a(t, X^u; u) := \nabla_{X_t}\left(\int_t^T \left(\frac{1}{2}\|u(X_t^u, t)\|^2 + f(X_t^u)\right) dt' + g(X_T^u)\right), \tag{89}$$

where $X^u$ solves (3). Then the dynamics of $a$ can be computed as

$$da_t = \left(\nabla_{X_t^u}(b(X_t^u, t) + \sigma(t)u(X_t^u, t))\right)^T a(t; X^u, u)dt$$
$$+ \left(\nabla_{X_t^u}(f(X_t^u, t) + \frac{1}{2}\|u(X_t^u, t)\|^2)\right) dt, \tag{90}$$
$$a(T; X^u, u) = \nabla g(X_T^u). \tag{91}$$

The path of $a$ is obtained by solving the above equations backwards in time given a trajectory $(X_t^u)$ and control $u$. Finally, the derivative of the relative entropy loss (88) is then computed as

$$\partial_\theta \mathcal{R} = \frac{1}{2}\int_0^T \frac{\partial}{\partial\theta}\|u(X_t^{\bar{u}}, t)\|^2 dt + \int_0^T \left(\frac{\partial u(X_t^{\bar{u}}, t)}{\partial\theta}\right)^T \sigma(t)a(t; X_t^{\bar{u}}, \bar{u})dt, \tag{92}$$

where the notation $\bar{u}$ means that we do stop gradients w.r.t. $\theta$ from flowing through these values. Both the discrete and continuous adjoint method have been shown to work well in practice (Nüsken & Richter, 2021; Bertsekas & Shreve, 1996; Domingo-Enrich, 2024). However, their training can be unstable due to the non-convexity of the problem.

**Variance and log-variance** For the second class of loss functions, let $(X_t^v)$ denote the solution to (3), with $u$ replaced by $v$. Then we can define

$$\widetilde{Y}_T^{u,v} = -\int_0^T (u \cdot v)(X_s^v, s)ds - \int_0^T f(X_s^v, s)ds - \int_0^T u(X_s^v, s) \cdot dW_s + \int_0^T \|u(X_s^v, s)\|^2 ds. \tag{93}$$

The *variance* and *log-variance* loss are then defined as

$$\mathcal{R}_{\text{Var}}(u) = \text{Var}(e^{\widetilde{Y}_T^{u,v} - g(X_T^v)}), \tag{94}$$
$$\mathcal{R}_{log-var}(u) = \text{Var}(\widetilde{Y}_T^{u,v} - g(X_T^v)). \tag{95}$$

It can be shown that these losses are minimized when $u = u^*$, irrespective of the choice of $v$. In Nüsken & Richter (2021), it was shown that these loss functions are closely connected to the FBSDE formulation of the SOC problem.

**SOCM** The Stochastic Optimal Control Matching (SOCM) loss, introduced in Domingo-Enrich et al. (2024b), is one of the most recently introduced IDO losses. It is described in the following theorem, where we adapt the notation $\lambda = \beta^{-1}$:

**Theorem 12** *(Domingo-Enrich et al., 2024b, Theorem 1) For each $t \in [0, T]$, let $M_t : [t, T] \to \mathbb{R}^{d \times d}$ be an arbitrary matrix-valued differentiable function such that $M_t(t) = \mathrm{Id}$. Let $v \in \mathcal{U}$ be an arbitrary control. Let $\mathcal{R}_{\mathrm{SOCM}} : L^2(\mathbb{R}^d \times [0, T]; \mathbb{R}^d) \times L^2([0, T]^2; \mathbb{R}^{d \times d}) \to \mathbb{R}$ be the loss function defined as*

$$\mathcal{R}_{\mathrm{SOCM}}(u, M) := \mathbb{E}\left[\frac{1}{T}\int_0^T \|u(X_t^v, t) - w(t, v, X^v, W, M_t)\|^2 \, dt \times \alpha(v, X^v, W)\right], \quad (96)$$

*where $X^v$ is the process controlled by $v$ (i.e., $dX_t^v = (b(X_t^v, t) + \sigma(t)v(X_t^v, t)) \, dt + \sqrt{\lambda}\sigma(t)dW_t$ and $X_0^v \sim p_0$), and*

$$w(t, v, X^v, W, M_t) = \sigma(t)^\top \Bigg( -\int_t^T M_t(s)\nabla_x f(X_s^v, s)ds - M_t(T)\nabla g(X_T^v)$$

$$+ \int_t^T \left(M_t(s)\nabla_x b(X_s^v, s) - \partial_s M_t(s)\right)(\sigma^{-1})^\top(s)v(X_s^v, s)ds$$

$$+ \lambda^{1/2}\int_t^T \left(M_t(s)\nabla_x b(X_s^v, s) - \partial_s M_t(s)\right)(\sigma^{-1})^\top(s)dW_s \Bigg),$$

$$\alpha(v, X^v, B) = \exp\Bigg( -\lambda^{-1}\int_0^T f(X_t^v, t)dt - \lambda^{-1}g(X_T^v)$$

$$- \lambda^{-1/2}\int_0^T \langle v(X_t^v, t), dW_t \rangle - \frac{\lambda^{-1}}{2}\int_0^T \|v(X_t^v, t)\|^2 dt \Bigg). \quad (97)$$

*$\mathcal{L}_{\mathrm{SOCM}}$ has a unique optimum $(u^*, M^*)$, where $u^*$ is the optimal control.*

The proof hinges on the path integral representation described in Kappen (2005a) and using a reparametrization trick to compute its gradients. We refer the reader to the relevant work Domingo-Enrich et al. (2024b) for more details. The reason why this method outperforms other IDO losses is as follows: the loss (96) can be seen as minimizing the discrepancy between $u$ and a target vector field $w$. The addition of the parametrized matrix $M$ allows the variance of this weight to be reduced, making it easier to learn. The downside of this method is that the variance of the importance weight $\alpha$ can blow up in more complex settings or with poor initialization, with the method failing to converge as a result.

**Adjoint Matching** The most recently introduced IDO loss, *adjoint matching*, was proposed in Domingo-Enrich et al. (2024a), and was proposed in the context of finetuning diffusion models. It is based on two observations. Firstly, one can write down a regression objective that does not have an importance weighting $\alpha$ by using the adjoint state $a$ defined earlier.

**Lemma 8** *(Domingo-Enrich et al., 2024a, Proposition 2) Define the basic adjoint matching objective as*

$$\mathcal{R}_{Basic-Adj-Match}(u; X^u) = \frac{1}{2}\int_0^T \|u(X_t, t) + \sigma(t)^T a(t; X^u, \bar{u})\|^2 dt, \quad \bar{u} = \texttt{stopgrad}(u). \quad (98)$$

*where $\bar{u} = \texttt{stopgrad}(u)$ means that the gradients of $\bar{u}$ w.r.t. the parameters $\theta$ of the control $u$ are artificially set to zero. Then the gradient of this loss w.r.t. $\theta$ is equal to (92), the gradient of the loss in the continuous adjoint method. Consequently, the only critical point of $\mathbb{E}_{X^u \sim \mathbb{P}^u}[\mathcal{R}_{Basic-Adj-Match}]$ is the optimal control $u^*$.*

Secondly, it is observed that some terms in the SDE for the adjoint state (90) have expectation zero under the trajectories of the optimal control. Indeed, it holds by definition of the value function and adjoint state that

$$\nabla V(x, t) = \mathbb{E}[a(t, X^{u^*}, u^*) \mid X_t = x] \quad (99)$$

and hence, since $u^* = -\sigma \nabla V$, we get

$$\mathbb{E}_{X \sim \mathbb{P}^{u^*}} \left[ u^*(x,t)^T \nabla_x u^*(x,t) + a(t, X, u^*)^T \sigma(t) u^*(x,t) \mid X_t = x \right] = 0 \qquad (100)$$

This motivates dropping the terms with expectation zero in (90), yielding the loss function

$$\mathcal{R}_{Adj-Match}(u; X) = \frac{1}{2} \int_0^T \| u(X_t, t) + \sigma(t)\tilde{a}(t; X, \bar{u}) \|^2 \mathrm{d}t, \quad \bar{u} = \texttt{stopgrad}(u), \qquad (101)$$

$$\text{where } \frac{\mathrm{d}}{\mathrm{d}t} \tilde{a}(t; X) = -(\tilde{a}(t; X)^T \nabla_x b(X_t, t) + \nabla_x f(X_t, t)), \qquad (102)$$

$$\tilde{a}(T, X) = \nabla_x g(X_T). \qquad (103)$$

The value $\tilde{a}$ is called the *lean adjoint state*. The claimed benefit of this method is that the resulting loss is a simple least-squares regression objective with no importance weighting, allowing it to avoid the problem of high variance importance weights. We refer to the original work Domingo-Enrich et al. (2024a) for a more in-depth discussion and proofs related to the adjoint matching loss.

## D ALGORITHMS

We give an overview of the algorithm used for eigenfunction learning in Algorithm 1, and for the IDO method in Algorithm 2.

---

**Algorithm 1** Deep learning for eigenfunctions

---

1: Parametrize the eigenfunctions $(\phi_i^{\theta_i})$, choose loss functions $(\mathcal{R}_i)$.
2: Fix a batch size $m$, number of iterations $N$, regularization $\alpha > 0$, learning rate $\eta > 0$.
3: Generate $m$ samples $(X_i)_{i=1}^m$ from $\mu$, for instance using an MCMC scheme.
4: **for** $n = 0, \ldots, N - 1$ **do**
5:     (optional) Update the samples $(X_i)_{i=1}^m$ using a sampling algorithm.
6:     Compute an $m$-sample Monte-Carlo estimate $\widehat{\mathcal{R}}_i(\phi_i^{\theta_i})$ of the loss $\mathcal{R}_i(\phi_i^{\theta_i})$ .
7:     Compute the gradients $\nabla_{\theta_i} \widehat{\mathcal{R}}_i(\phi_i^{\theta_i})$ for the current parameters $\theta_i$, and update $\theta_i$ using Adam.
8: **end for**
9: Estimate the eigenvalues $\lambda_i$ from the learned functions $\phi_i^{\theta_i}$
10: **return** eigenfunction estimates $(\phi_i^{\theta_i})$, eigenvalue estimates $(\lambda_i)$.

---

---

**Algorithm 2** Iterative Diffusion Optimization (IDO)

---

1: Parametrize the control $u_\theta \in \mathcal{U}, \theta \in \mathbb{R}^p$, and choose a loss function $\mathcal{R}$.
2: Fix a batch size $m$, number of iterations $N$, number of timesteps $K$, learning rate $\eta > 0$.
3: **for** $i = 0, \ldots, N - 1$ **do**
4:     Simulate $m$ trajectories of (3) with control $u = u_\theta$ using a $K$-step discretization scheme.
5:     Compute an $m$-sample Monte-Carlo estimate $\widehat{\mathcal{R}}(u_\theta)$ of the loss $\mathcal{R}(u_\theta)$.
6:     Compute the gradients $\nabla_\theta \widehat{\mathcal{R}}(u_\theta)$ for the current parameters $\theta$ and update using Adam.
7: **end for**
8: **return the learned control** $u_\theta \approx u^*$.

---

# E EXPERIMENTAL DETAILS

## E.1 SETTING CONFIGURATIONS

To evaluate the learned controls, we will use three different metrics:

1. The *control objective* is the value we are trying to minimize,

$$\mathbb{E}\left[\int_0^T \left(\frac{1}{2}\|u(X_t^u, t)\|^2 + f(X_t^u)\right) \mathrm{d}t + g(X_T^u)\right]. \tag{104}$$

   It can be estimated by simulating trajectories of (3). Unless mentioned otherwise, we estimate it using 65536 trajectories and report the standard deviation of the estimate.

2. The *control $L^2$ error at time $t$* is given by

$$\mathbb{E}_{x \sim \mathbb{P}_t^{u^*}}\left[\|u(x, t) - u^*(x, t)\|^2\right], \tag{105}$$

   where $\mathbb{P}_t^{u^*}$ denotes the density over $\mathbb{R}^d$ induced by the trajectory (3) under the optimal control at time $t$.

3. The *average control $L^2$ error* is the above quantity averaged over the entire trajectory,

$$\mathbb{E}_{t \sim [0,T]}\mathbb{E}_{x \sim \mathbb{P}_t^{u^*}}\left[\|u(x, t) - u^*(x, t)\|^2\right]. \tag{106}$$

These are also the metrics reported in previous works Domingo-Enrich et al. (2024b); Domingo-Enrich (2024); Nüsken & Richter (2021). The $L^2$ error is introduced because, after the control $u$ is sufficiently close to the optimal control $u^*$, the expectation (104) requires increasingly more Monte Carlo samples to distinguish $u$ from the optimal control. In this case the $L^2$ error is a more precise metric for determining how close a given control is to the optimal control.

For all experiments, we trained the neural networks involved using Adam with learning rate $\eta = 10^{-4}$. For the IDO methods, we use a batch size of $m = 64$. For the eigenfunction learning, we sample from $\mu$ using the Metropolis-Adjusted Langevin Algorithm (Roberts & Rosenthal, 1998). Unless mentioned otherwise, we sample $m = 65536$ samples, updating these in every iteration with 100 MCMC steps with a timestep size of $\Delta t = 0.01$ after a warm-up phase of 1000 steps. For more details, we refer to the code in the supplementary material, which contains a description of all hyperparameters used.

QUADRATIC    The first setting we consider has

$$E(x) = \frac{1}{2}x^T A x, \quad f(x) = x^T P x, \quad g(x) = x^T Q x, \tag{107}$$

where $A \in \mathbb{R}^{d \times d}$ is symmetric, $Q \in \mathbb{R}^{d \times d}$ is positive definite and $P \in \mathbb{R}^{d \times d}$. This type of control problem is more widely known as the linear quadratic regulator. The optimal control is given by $u^*(x, t) = -2F_t x$, where $F_t$ solves the Riccati equation (see (van Handel, 2007, Theorem 6.5.1))

$$\frac{\mathrm{d}F_t}{\mathrm{d}t} - A^T F_t - F_t A - 2F_t F_t^T + P = 0, \quad F_T = Q. \tag{108}$$

We consider three different configurations:

- (ISOTROPIC) We set $d = 20$, $A = I$, $P = I$, $Q = 0.5I$, $\beta = 1$, $T = 4$, $x_0 \sim \mathcal{N}(0, 0.5I)$, taking $K = 200$ time discretization steps for the simulation of the SDE.

- (REPULSIVE) Exactly the same as isotropic, but with $A = -I$.

- (ANISOTROPIC) We set $d = 20$, $A = \mathrm{diag}(e^{a_i})$, $P = U\mathrm{diag}(e^{p_i})U^T$, $Q = 0.5I$, $\beta = 1$, $T = 4$, $x_0 \sim \mathcal{N}(0, 0.5I)$, taking $K = 200$ time discretization steps for the simulation of the SDE. The values $a_i, p_i$ are sampled i.i.d. from $\mathcal{N}(0, 1)$, and the matrix $U$ is a random orthogonal matrix sampled using `scipy.stats.ortho_group` (Virtanen et al., 2020) at the start of the simulation.

DOUBLE WELL   The second setting we consider is the $d$-dimensional double well defined through

$$E(x) = \sum_{i=1}^{d} \kappa_i (x_i^2 - 1)^2, \quad f(x) = \sum_{i=1}^{d} \nu_i (x_i^2 - 1)^2, \quad g(x) = 0. \tag{109}$$

where $d = 10$, and $\kappa_i = 5, \nu_i = 3$ for $i = 1, 2, 3$ and $\kappa_i = \nu_i = 1$ for $i \geq 4$. In addition, we again set $T = 4$, $\beta = 1$ and use $K = 400$ discretization steps. This problem is similar to the one considered in Nüsken & Richter (2021) and Domingo-Enrich et al. (2024b), the difference being that we consider a longer horizon ($T = 4$ instead of $T = 1$) and consider a nonzero running cost in order to have nontrivial long-term behaviour. This problem is considered a highly nontrivial benchmark problem, since the double well in each dimension creates a total of $2^d = 1024$ local minima, making this setting highly multimodal. The ground truth is *not* available in closed form, but can be approximated efficiently by noticing that the energy and running cost are a sum of one-dimensional terms, and hence we can compute the solution by solving $d$ one-dimensional problems using a classical solver.

RING   The final setting considers the setup

$$E(x) = \alpha \left( \|x\|^2 - 2R^2 \right) \|x\|^2, \quad f(x) = 2x_1, \quad g(x) = 0, \tag{110}$$

where $\alpha = 1, R = 5/\sqrt{2}$. This energy is nonconvex and has its minimizers lying on the hypersphere with radius $R$. This setting serves to highlight the difference between the relative eigenfunction loss (20) and the other eigenfunction losses, and for visualization purposes will be done in $d = 2$. In addition, we set $T = 5, \beta = 1$ and $x_0 = (R, 0)$. The energy landscape and running cost are visualized in Figure 6. The goal of the controller is essentially to guide the system to the left hand side of the $xy$-plane while being constrained by the potential to stay close to the circle of radius $R$. This setting requires a smaller timestep for stable simulation, we take $K = 500$.

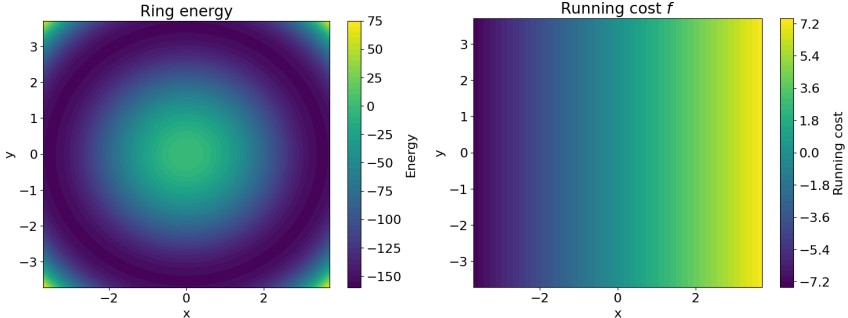

Figure 6: Energy function (left) and running cost (right) for the RING setting in $d = 2$.

### E.2 MODEL ARCHITECTURE AND TRAINING

**Architecture**   For the IDO methods, we use the exact same architecture as in Domingo-Enrich et al. (2024b). They argue that the control can be viewed as the analog of a score function in diffusion models, and hence they use a simplified U-Net architecture, where each of the up-/downsampling steps is a fully connected layer with ReLU activations. As in their work, we use three downsampling and upsampling steps, with widths 256, 128 and 64.

For the eigenfunction models, we use the same architecture, but replace the ReLU activation with the GELU activation function $x \mapsto x\Phi(x)$, where $\Phi$ is the cdf of a standard normal distribution (Hendrycks & Gimpel, 2023). This is done because the eigenfunction losses require evaluating the derivatives of the network w.r.t. the inputs, hence requiring a smoother activation function.

**Training**   For the eigenfunction method, we train using the following procedure:

1. Start training the top eigenfunction using the deep Ritz loss (17). Every 100 iterations, we estimate the eigenvalue $\lambda_0$, and continue training until the variance of these estimates (computed with EMA 0.5) is below $10^{-4}$ and we have reached at least 5000 iterations.

Table 2: Iteration times by method and loss

| Method | Experiment Loss | Iteration time ($s$) |
|---|---|---|
| **COMBINED** | **Adjoint Matching** | 0.332 |
| | **Log variance** | 0.328 |
| | **Relative entropy** | 0.419 |
| | **SOCM** | 0.432 |
| **EIGF** | **Deep ritz loss** | 0.227 |
| | **PINN** | 0.662 |
| | **Relative loss** | 0.662 |
| | **Variational loss** | 0.228 |
| **FBSDE** | **FBSDE** | 0.443 |
| **IDO** | **Adjoint Matching** | 0.230 |
| | **Log variance** | 0.212 |
| | **Relative entropy** | 0.413 |
| | **SOCM** | 0.799 |

2. Fix $\lambda_0$, and continue training the top eigenfunction using a loss function of choice among (15),(17), (20). Start training the excited state using the variational loss $\mathcal{R}_{\mathrm{Var}}$ in (73) with $k = 1$ and regularization parameter $\alpha = |\lambda_0|$.

For the combined methods, we first train the eigenfunction and eigenvalues using the method above with $\mathcal{R}_{Rel}$ for 80 000 iterations, and then start training with an IDO loss using the control parametrization (22). We resample the trajectories in $[0, T_{cut}]$ (which only use the eigenfunction control) every $L = 100$ iterations in order to have a diverse set of starting positions at $T = T_{cut}$.

### E.3 COMPUTATIONAL COST

Table 2 shows the computation cost per iteration for the different algorithms for the QUADRATIC (REPULSIVE) setting, measured in seconds/iteration when ran in isolation on a single GPU. All experiments where carried out on an NVIDIA H100 NV GPU.

### E.4 FURTHER DETAILS ON THE RING SETTING

The RING setting serves as an illustrative example the difference between the 'absolute' eigenfunction losses and our relative eigenfunction loss. As shown in Figure 4, the relative loss obtains a drastically lower control objective than the absolute losses. To further understand this, consider the shape of the learned eigenfunctions for the different losses, shown in Figure 7. From the top row, it is clear that the learned eigenfunctions for the different methods are all very close in terms of the distance induced by $\| \cdot \|_\mu$. However, while the learned eigenfunctions look similar in $L^2(\mu)$, the logarithm of the learned eigenfunctions varies drastically, and hence the resulting control $\nabla \log \phi$ (shown in Figure 3) differs greatly - the control learned by the relative loss correctly guides the system along the circle in the negative $x$ direction, while the other controls are not learned correctly for $x \geq 0$. The name 'relative loss' comes from the analogy with *absolute* and *relative* errors,

$$|x - x^*| < \epsilon \quad \text{vs.} \quad \left| \frac{x}{x^*} - 1 \right| < \epsilon \iff |\log x - \log x^*| < \epsilon + \mathcal{O}(\epsilon^2). \tag{111}$$

In essence, since the resulting control is given by $\nabla \log \phi$, the quantity of interest is the *relative* error of the learned eigenfunction, while existing methods are designed to minimize the *absolute* error.

### E.5 DETAILS ON FIGURE 1

We give here some details on the motivating plot shown in Figure 1. The value shown is the control $L^2$ error of each of the different methods in the QUADRATIC (REPULSIVE) setting. Each algorithm was run for 30k iterations, and the values reported are the mean and 5%-95% quantiles over the last

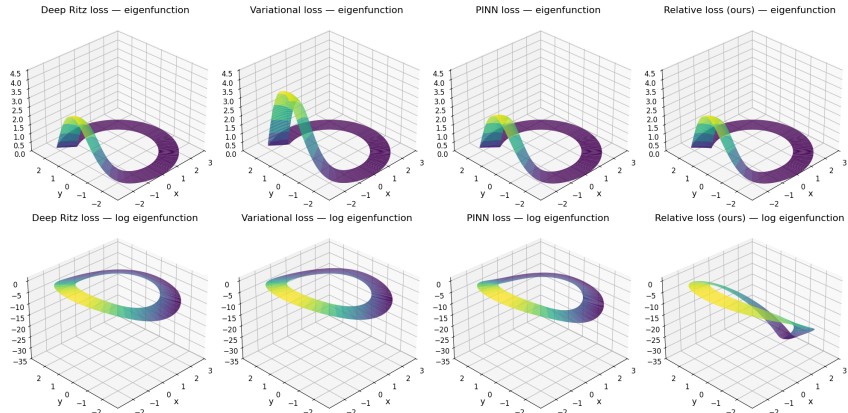

Figure 7: Learned eigenfunctions (top row) and their logarithms (bottom row) for the RING setting with different loss functions.

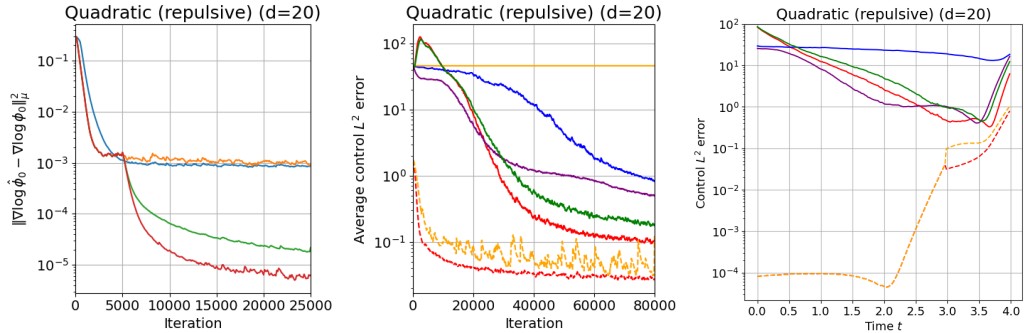

Figure 8: $L^2$ error of $\nabla \log \phi$ for different eigenfunctions (left), average control $L^2$ error for different methods (middle) and control $L^2$ error over time (right). Legend is the same as in Figure 4 and Figure 5.

1000 iterations. The EIGF+IDO (OURS) label refers to the combined method, where we first train the eigenfunctions until convergence and then train using the relative entropy loss.

### E.6 ADDITIONAL EXPERIMENT: REPULSIVE POTENTIAL

Figure 8 shows the same plots discussed in the main text for the QUADRATIC (REPULSIVE) setting, where we obtain similar results. The reported eigenfunction error is measured in $L^2(\overline{\mu})$ instead of $L^2(\mu)$.

Additionally, for this experiment setting we have tested the performance of "ergodic" control estimator. It is based only on the first eigenfunction, i.e. $\beta^{-1} \nabla \log \phi_0^{\theta_0}$ is used for the whole time range. The performance of such an estimator is presented in Figure 9 by a curve labeled EIGF. It can be seen, that this approach reduces the error down with growth of time horizon $T$. However, using our proposed approach with control as in (22), which corresponds to EIGF+IDO curve, leads to a significant improvement.

### E.7 CONTROL OBJECTIVES

As mentioned before, the control objective is the final metric for evaluating the performance of the different algorithms, but due to variance of the Monte Carlo estimator the difference between methods can be quite small. We report the control objectives for all experiments at convergence in Table 3 and Table 4. The value reported is the mean value of the control objective over $N = 65536$ simulations, and the error is the standard deviation of these estimates, divided by $\sqrt{N}$.

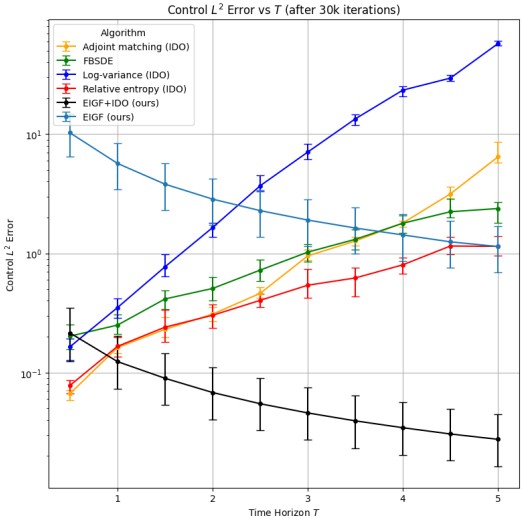

Figure 9: In this experiment EIGF method uses an ergodic estimator based only on the first eigenfunction. EIFG+IDO curve corresponds to the application of the proposed controller (22) with Relative Entropy loss. The figure shows $L^2$ control error for different methods after 30000 iterations.

Table 3: Control objective for the different methods in the QUADRATIC (ISOTROPIC) and QUADRATIC (REPULSIVE) settings. The SOCM method did not converge, and hence the dynamics diverge.

| Method | Loss | QUADRATIC (ISOTROPIC) | QUADRATIC (REPULSIVE) |
|---|---|---|---|
| **IDO** | **Relative entropy** | $32.7870 \pm 0.014$ | $112.5172 \pm 0.051$ |
| | **Log variance** | $32.7850 \pm 0.014$ | $114.0552 \pm 0.053$ |
| | **SOCM** | $73.1062 \pm 0.062$ | $\text{nan} \pm \text{nan}$ |
| | **Adjoint matching** | $32.7754 \pm 0.014$ | $113.4554 \pm 0.052$ |
| **COMBINED** | **Relative entropy** | $32.7763 \pm 0.014$ | $\mathbf{112.3444 \pm 0.050}$ |
| **(ours)** | **Log variance** | $32.7740 \pm 0.014$ | $150.0721 \pm 0.12$ |
| | **SOCM** | $\mathbf{32.7717 \pm 0.014}$ | $112.3960 \pm 0.050$ |
| | **Adjoint matching** | $32.7725 \pm 0.014$ | $114.7020 \pm 0.055$ |
| **FBSDE** | **FBSDE** | $32.7979 \pm 0.014$ | $112.6393 \pm 0.051$ |

Table 4: Control objective for the different methods in the QUADRATIC (ANISOTROPIC) and DOUBLE WELL settings.

| Method | Loss | QUADRATIC (ANISOTROPIC) | DOUBLE WELL |
|---|---|---|---|
| **IDO** | **Relative entropy** | $38.9967 \pm 0.022$ | $35.2688 \pm 0.010$ |
| | **Log variance** | $31.3664 \pm 0.016$ | $32.8645 \pm 0.0094$ |
| | **SOCM** | $112.2728 \pm 0.18$ | $41.7215 \pm 0.013$ |
| | **Adjoint matching** | $31.3584 \pm 0.016$ | $34.8713 \pm 0.010$ |
| **COMBINED** | **Relative entropy** | $\mathbf{31.3476 \pm 0.016}$ | $32.6130 \pm 0.0088$ |
| **(ours)** | **Log variance** | $32.5115 \pm 0.047$ | $32.9080 \pm 0.0088$ |
| | **SOCM** | $31.3483 \pm 0.016$ | $\mathbf{32.4421 \pm 0.0088}$ |
| | **Adjoint matching** | $31.3497 \pm 0.016$ | $32.5638 \pm 0.0088$ |
| **FBSDE** | **FBSDE** | $31.3854 \pm 0.016$ | $35.1890 \pm 0.011$ |

