# OpenReview forum: "A Schrödinger Eigenfunction Method for Long-Horizon Stochastic Optimal Control"
_ICLR.cc/2026/Conference — ICLR 2026 Poster_

### Official Review · Reviewer_Y3as · 2025-10-28

**Soundness:** 4
**Presentation:** 4
**Contribution:** 4
**Rating:** 8
**Confidence:** 4

**Summary:**

This paper studies the numerical resolution of continuous control PDEs (a.k.a. HJB equations) in high-dimensional state spaces using neural networks (as classical PDE solvers are prohibited by the curse of dimensionality). Contrary to most works, which focus on the computational impact of the number of samples of the trajectories, here the concern is the computational impact of the length of the trajectories. In this undiscounted, finite, but large horizon problem, the authors propose expanding the evolution operator into its eigenfunctions and keeping only the dominant eigenfunction as a numerical solution. This raises three sub-problems: 1) identifying conditions under which this approach is principled, 2) identifying or approximating the principal eigenfunction, 3) recovering an effective control from the principal eigenfunction.

The authors tackle 1) by working within a classical, moderate-complexity setting of affine control, whose (thus simplified) evolution operator can be related through a classical change of function to a linear PDE and associated Schrödinger operator. These theoretical derivations follow classical methodologies in control and PDE theory, but their highly qualitative and rigourous application yields novel results, and in particular Theorem 4. They also naturally yield an expression of the optimal control in terms of the eigenbasis (Theorem 3), which is truncated to provide a solution to sub-problem 3). This analytical method is the main body of the work, an architecture, if you will, which supports the use of neural network methods for learning eigenfunctions to solve sub-problem 2). The authors provide an extensive overview of existing methods for learning eigenfunctions, but also propose their own loss, which is expressed in units corresponding to the control problem (a reweighting which counteracts the change of functions of step 1)).

**Strengths:**

The paper is well-written, well-structured, and highly pedagogical, despite the difficulty of the topic and the distance between the communities of PDE theory and general ML. It is also very complete both in terms of the references in control and PDE theories and in terms of the rigour of the statements and argumentation. The theoretical contributions are impressive and support a novel multi-disciplinary approach fusing control/PDEs with ML, while being backed up by very rigourous mathematics and good experiments.

**Weaknesses:**

The assumptions required to obtain the theoretical results are more general than, say, linear quadratic, but they remain a far cry from many practical situations. This is highlighted in the experiments, which are restricted to toy problems. However, in my opinion, while it is a weakness, it is difficult to criticise the quality of the work on these grounds in good faith, as the step up in complexity required by weaker assumptions would be so vast.

_Minor feedbacks_:
- The later appendices (C and beyond) contain a few typos and overloaded notations; there are also some less-than-correct phrases in the main text.
- In equation (104), shouldn’t the integral over $t$ be a classical integral and not an expectation?

**Questions:**

There is one important question that, for me, casts a shadow over the phenomenon at play here, which I think ought to be answered:
- If the horizon is becoming longer and longer, one would expect turnpike properties to kick in and the ergodic regime to become a good approximation for the behaviour of the system. This connection is suggested in Remarks 1 and 2. To what degree is this actually responsible for the improvement in computational performance? Wouldn’t a good benchmark for Figure $1$ be a solver which solves the ergodic version of the problem (though you would have to change the problem to one of the other LQRs for this to exist), which one would expect to improve with $T$ just by the law of large numbers?


This work has made me wonder about several research questions, which is something I like about it, but these wouldn’t change my evaluation of the work. I leave them without expecting an answer for the authors’ benefit:
- You use only the first eigenvalue here, but you could consider higher-order approximations, which might improve performance. This performance would be inversely exponential in the spectral gap, of course, so it would be an interesting question to study the gaps in the spectra of different evolution operators to identify problems where different orders of approximation are interesting.
- If you could study the full HJB equation (beyond control-affine assumptions, I mean) and show that the evolution operator has a spectral gap and eigenbasis, you could apply the learning components as well, no? Perhaps using hypocoercivity theory and some non-linear PDE analysis, it would be possible to obtain similar results under very general assumptions, though it would be very complicated.
- I understand the problem with end-of-the-world effects, which cause the need to switch off the eigenfunction solver near $T$, but the performance degradation is very painful (Figure 5). Perhaps something clever can be done about it.

---

> ### Author Response · Authors · 2025-11-25
>
> We thank the reviewer for the careful and thoughtful examination of our work. We respond to the comments below.
>
> **Response to Minor feedback.** We would like to thank the reviewer for providing a useful feedback. We will give another look to the Appendix section and search for possible misprints. Additionally, we agree that a more classical metric with the integral over $t$ in equation (104) might be used as, for example, done in Nusken and Richter (2021). However, we conducted the experiments using the expectation form, which should be equivalent for experiments with the same time horizon.
>
> **Ergodic limit as a solution approximation.** The solution for the ergodic problem would yield a similar controller to ours, which only uses the first eigenfunction, as suggested by Eq. (2). We have conducted an experiment using the proposed ''ergodic'' control estimator, please see Appendix E.8 and Figure 9 for details in the revised version of the manuscript. We can see that for a larger horizon we indeed tend to get a smaller $L^2$ error with ''ergodic'' controller, but using the full controller as in Eq. (2) yields smaller error for earlier times compared to the ''ergodic'' one.
>
> We want to note, that although the experiment in Figure 9 is conducted for dynamics with non-confining energy, we still are able to provide ergodic control estimator. This is due to the fact, that although the underlying dynamics have non-confining potential, the controlled dynamics exhibit ergodic behavior. Moreover, our operator-based approach can capture under which conditions this situation is possible in theory as well as provides a neural solution expressed via the first eigenfunction, which opens the door to other applications.
>
> **Research questions.** We would like to thank the reviewer for coming up with interesting the research questions. They look very exciting, but require more thought, work, and time - much more than available for the rebuttal period. Thus we would have to leave them for further work.

---

> > ### Comment · Reviewer_Y3as · 2025-11-26
> >
> > Thank you for the clear and insightful response.
> >
> > Addressing the relation to the ergodic controller was my main concern, which the authors have done.
> > I haven't made any changes to my review.

---

### Official Review · Reviewer_KCj8 · 2025-10-31

**Soundness:** 3
**Presentation:** 3
**Contribution:** 3
**Rating:** 6
**Confidence:** 3

**Summary:**

The authors present a **hybrid framework** for solving **stochastic optimal control (SOC)** problems under the **gradient-drift assumption**, combining **eigenfunctions** with **short-horizon solvers**. Using the **Cole–Hopf transform**, the **Hamilton–Jacobi–Bellman (HJB)** equation is converted into a **linear PDE**, enabling efficient solution through **eigenvalue and spectral analysis**. The framework reparameterizes the value function to learn the **top eigenfunction**, which determines the long-horizon control. Empirically, the method achieves strong performance for **long time horizons**, where existing approaches typically suffer from rapid error and computational cost growth.

**Strengths:**

1. Rigorous theoretical treatment with clear assumptions, theorems, and proofs.
2. Empirical results demonstrate improved performance over baselines as the horizon T increases.

**Weaknesses:**

1. The paper could better **motivate the relevance and generality** of the gradient-drift assumption, clarifying how broad and practically applicable the class of SOC problems is where the HJB can be reduced to a linear PDE.
2. The claim that “the exponential decay of the correction term in Eq. (2) suggests that the optimal control ($u^*$) can be approximated by the gradient of the logarithm of the top eigenfunction” holds only **asymptotically**; the bounded correction term may remain **non-negligible** over ([0, T]), especially at earlier times.
3. The modification of the loss from Eq. (15) to Eq. (20) is said to be equivalent when ($\phi$) is unit-norm, but since the regularization coefficient ($\alpha$) is **chosen heuristically**, there is **no guarantee** that both objectives share the same optimum or yield a normalized eigenfunction.
4. The authors state that the loss in Eq. (21) remains sensitive when ($\exp(-\beta V_0)$) is small; however, this may also cause **instability**, as the loss could **diverge** when ($V_0$) becomes large.

**Questions:**

1. How is the cutoff time ($T_{\text{cut}}$) selected or tuned in practice?
2. Does the learned eigenfunction ($\phi_\theta$) correspond to the true eigenfunction of the operator (L), or is it only an approximation under the chosen loss?
3. Is the eigenvalue ($\lambda_0$) learned during training (fine-tuning)? If so, is its computation **differentiable** and integrated into the optimization process?

---

> ### Author Response · Authors · 2025-11-25
>
> We appreciate the reviewer’s comments and address them point-by-point below.
>
> Response to the "Weaknesses" section:
>
> **Response to Comment 1: Gradient-drift assumption.** We agree with the reviewer that the motivation for the gradient–drift assumption could be made clearer. In the manuscript, we cite several references illustrating real-world settings where gradient drift naturally arises, including overdamped molecular dynamics, mean-field games, the control of particles interacting via potentials, and social models of opinion formation. These are widely used modeling frameworks in which gradient drift is not an artificial simplification but a standard, ground-truth description of the underlying processes.
>
>
> **Response to Comment 2: Asymptotic estimator behavior.** We agree with the reviewer that the behavior of the correction term at earlier times is not clear from Eq. (2). A more detailed analysis would require refined information about all eigenpairs and a deeper examination of Eq. (12), which lies outside the scope of this work. However, we do account for this issue in our practical procedure. The correction term is learned explicitly to capture its time dependence. Therefore, although this behavior is not immediately evident from the form of Eq.~(2), we believe that the learned correction term accurately reflects the contribution of the full series in Eq. (12), regardless of whether this contribution is negligible or, conversely, significant at earlier times.
>
>
> **Response to Comment 3: Optimum for related loss functions.** We agree with the reviewer that the behavior of the relative loss function and the characterization of its minimizers require further understanding. In contrast, for the variational loss we do have a clear description of the minimizer set, as shown in Appendix C.2: the minimizers are rescaled eigenfunctions, with the constant $\alpha$ appearing explicitly in the characterization. Our belief is that, at least at a functional level, the relative loss admits a similar set of minimizers, and that an appropriate normalization constant can subsequently be recovered via a scalar product. This would bridge the conceptual gap, but we do not yet have a rigorous justification and therefore leave a full analysis of this question to future work.
>
>
> **Response to Comment 4: Sensitivity of the loss and instability.** We agree with the reviewer that care is required here, and that one must balance the trade-off described in the reviewer's comment. We would like to note that the neural network models used in our experiments, together with our choice of activation functions, exhibit at most polynomial growth. For this reason, we believe that gradient-based optimization and regularization of the loss in Eq. (21) and of the model $V_0$ itself is more tractable than optimizing losses in which $V_0$ appears inside an exponential, where instability issues tend to be more difficult to control.

---

> > ### Author Response · Authors · 2025-11-25
> >
> > Response to the "Questions" section:
> >
> > **Response to Question 1: Tuning the cutoff time.**
> > In practice, $T_{\mathrm{cut}}$ is treated as a hyperparameter that is selected separately for each application. In brief, one must balance the trade–off between computational cost and performance. We discuss this in more detail in our response to **Question 4** from reviewer AKir, as well as provide a practical heuristic that typically yields a good value of $T_{\mathrm{cut}}$.
> >
> > **Response to Question 2: Learned eigenfunction.**
> > The minimizer of the variational loss in Eq. (17) is the true
> > eigenfunction $\phi_0$. In contrast, for the relative loss in Eq. (20),
> > the eigenvalue $\lambda_0$ must be estimated. In case if ground-truth value for $\lambda_0$ is used, we believe that the minimizer of the relative loss yields true eigenfunction under appropriately chosen functional class supported by Theorem 2. In case, if approximation is used for $\lambda_0$, the same
> > guarantee does not apply. Providing a theoretical bound on the
> > approximation error associated with the relative loss is beyond the
> > scope of this work. Nevertheless, our procedure first obtains an
> > initialization of the eigenfunction using the variational loss in
> > Eq. (17), which provides a reliable estimate of the true eigenfunction.
> > This initialization helps avoid potential approximation errors that
> > could arise if one attempted to learn directly using the relative loss
> > in Eq. (20) from the start.
> >
> > Some additional intuition may be gained from the following observation.
> > For the PINN loss weighted by $\mu$, a small perturbation in the
> > eigenvalue does not change the identity of the minimizer. Writing
> > $\phi = \sum_i \alpha_i \phi_i$ with $\sum_i |\alpha_i|^2 = 1$, we have
> >
> > $$
> > \|\mathcal{L}\phi - \lambda \phi\|_{\mu}^2
> > = \sum_i |\alpha_i|^2 (\lambda_i - \lambda)^2,
> > $$
> >
> > which is minimized at $\alpha_0 = 1$, $\alpha_{i>0} = 0$ whenever
> > $|\lambda - \lambda_0| < |\lambda - \lambda_1|$. We hope that the same property might hold for the relative loss as well. However, we are not
> > aware of a rigorous extension of this argument to the relative-loss
> > setting, and we leave this question for future work.
> >
> >
> > **Response to Question 3: Computation of $\lambda_0$.** The value of $\lambda_0$ is estimated during the first stage of training, and is *not* altered during the fine-tuning stage (it is not a learnable parameter by itself). It is estimated based on the known minimizer of the variational loss (see Appendix C.3).

---

### Official Review · Reviewer_AKir · 2025-11-09

**Soundness:** 3
**Presentation:** 2
**Contribution:** 3
**Rating:** 4
**Confidence:** 4

**Summary:**

Within the context of stochastic optimal control, this paper shows that optimal control can be obtained from the eigensystem of a Schrödinger operator S=−\Delta + V with purely discrete spectrum. The paper also states that for a symmetric linear-quadratic regulator (LQR), the S matches the Hamiltonian of a quantum harmonic oscillator, whose closed-form eigensystem yields an analytic solution to the symmetric LQR with arbitrary terminal cost. For the general version, the paper presents a learning strategy of the eigensystem of the operator L via physics informed neural  networks.

**Strengths:**

- An interesting connection between a version of stochastic optimal control and the Schrodinger operator.
- A deep learning strategy based on PINN and several loss functions to learn the eigensystem of the operator L.

**Weaknesses:**

-  The presentation of the theoretical results is a bit unclear. For example, it is not clear what is the different between the presented Theorem 2 and the (Reed & Simon, 1978, Theorem XIII.67, XIII.47. Appendix B.5 announced to provide a proof of Theorem 2, but that is hardly the case, it is at best a sketch and a restatement of the results in (Reed & Simon, 1978, Theorem XIII.67, XIII.47.
- In the numerical results, how is the check that the eigenvalues do not have a finite accumulation point?
- The paper states that Instead of learning V_0 and \lambda_0 jointly, it first performs training
with a variational loss in equation (17) to obtain an estimate for the eigenvalue \lambda_0 and an initialization of V_0, and then ‘fine-tune’ using equation (20). Can the authors comment on the convergence rate? What affects the convergence as a function of dimensionality d?
- Another issue is related to how effective is the proposed eigensystem learning approach when addressing the long-horizon challenge? Although the paper states under the limitations that method for determining an appropriate cutoff time exist, can computational approaches shade some light on the T_cut?
- When discussing order-of-magnitude, what sustains the improvement in L^2 error compared to standard IDO and FBSDE methods? Some additional explanations could have help me understand also the computational complexity analysis.
- Although this is a minor issue, when considering the cases without gradient drift (where L is non-symmetric), what are the major technical problems?

**Questions:**

-  The presentation of the theoretical results is a bit unclear. For example, it is not clear what is the different between the presented Theorem 2 and the (Reed & Simon, 1978, Theorem XIII.67, XIII.47. Appendix B.5 announced to provide a proof of Theorem 2, but that is hardly the case, it is at best a sketch and a restatement of the results in (Reed & Simon, 1978, Theorem XIII.67, XIII.47.
- In the numerical results, how is the check that the eigenvalues do not have a finite accumulation point?
- The paper states that Instead of learning V_0 and \lambda_0 jointly, it first performs training
with a variational loss in equation (17) to obtain an estimate for the eigenvalue \lambda_0 and an initialization of V_0, and then ‘fine-tune’ using equation (20). Can the authors comment on the convergence rate? What affects the convergence as a function of dimensionality d?
- Another issue is related to how effective is the proposed eigensystem learning approach when addressing the long-horizon challenge? Although the paper states under the limitations that method for determining an appropriate cutoff time exist, can computational approaches shade some light on the T_cut?

**Details Of Ethics Concerns:**

None detected

---

> ### Author Response · Authors · 2025-11-24
>
> We would like to thank the reviewer for engaging with the work and raising several points that we address in detail below.
>
> Reply, part 1 out of 2:
>
> **Response to Question 1: Appendix B.5 and Proof of Theorem 2.**
> We would like to clarify that Theorem 2 is **not** intended as a new theoretical contribution. Its purpose is to collect and summarize existing results that form the theoretical foundation of our spectral method.
> Since all statements in Theorem 2 are known, its proof consists of restating and adapting relevant results from Reed and Simon (1978). We therefore agree with the Reviewer on this point and will make this clearer in the revised manuscript.
>
> **Response to Question 2: Finite accumulation point of the operator.**
> In our setting, the absence of finite accumulation points of the eigenvalues is a theoretical property guaranteed by Theorem 2. Consequently, there is no need to check this numerically, and our experiments do not rely on such a verification. We have also revised the statement of Theorem 6 to make the role of this property more explicit. Please see the updated manuscript for clarification.
>
> **Response to Question 3: Convergence rate.**
> Our parametrization reduces the problem to minimizing the loss functions in Eq. (19) for estimating $\lambda_0$ and $V_0$, followed by fine-tuning using the loss in Eq. (21). Since all these objectives are optimized with respect to deep neural-network parameters, the resulting problems are inherently non-convex, and a rigorous theoretical convergence analysis is generally intractable in this setting. For this reason, providing a formal convergence rate is beyond the scope of this work. Similarly, the dependence of the convergence behavior on the dimension $d$ is not theoretically characterized at present. Empirically, the difference in convergence rate behavior between different loss functions was shown in Figure 4.
>
> We provide additional results to empirically address the reviewer's question about the dependence of the convergence rate on the problem dimension. While it is difficult to characterize this dependence by a single explicit scalar rate, we believe a sensible empirical characterization is the following: for different dimensions \(d\), we run our method for a fixed number of iterations and then compare the resulting final error as well as the error-per-dimension across dimensions. We present the additional results for the same experimental setting as in the leftmost panel in Figure 4: Quadratic Isotropic case, where training was done using variational loss until iteration 5000, and after iteration 5000 using relative loss until iteration 20000. Results are displayed in the following table:
>
> | dimension | grad_log_error | grad_log_error_over_dim |
> |-----------|-----------------|--------------------------|
> | 4         | 0.007113        | 0.001778                 |
> | 8         | 0.009439        | 0.00118                  |
> | 12        | 0.01053         | 0.0008773                |
> | 16        | 0.01537         | 0.0009606                |
> | 20        | 0.01574         | 0.0007872                |
> | 24        | 0.01883         | 0.0007847                |
>
>
> **Response to Question 4: Long horizon and methods to determine cutoff time.**
> The motivation of our work is exactly to handle long-horizon settings. And we indeed see that the provided eigensystem approach to be effective in terms of $L^2$ error as shown in Figure 5. However, we are not sure whether the reviewer refers to this result in the first question. We would be very grateful if the reviewer could further clarify his question.
>
> Regarding the choice of the cutoff time $T_{cut}$, we have the following intuition.
> Increasing $T_{\mathrm{cut}} \to T$ raises computational cost, whereas
> choosing $T_{\mathrm{cut}}$ too small degrades performance; this reflects
> the fundamental trade-off in selecting this parameter. One heuristic for
> choosing an appropriate order of magnitude is to observe that we would
> like the correction term given by the infinite series in Eq. (12) to be
> small for all $t \leq T_{\mathrm{cut}}$. Even without access to the inner
> products or eigenfunctions, we may consider the approximation
> $$
> \exp\left(-\tfrac{1}{2\beta}(\lambda_1 - \lambda_0)(T - T_{\mathrm{cut}})\right)
> = \varepsilon,
> $$
> which yields
> $$
> T - T_{\mathrm{cut}}
> = -\frac{2\beta}{\lambda_1 - \lambda_0}\,\log \varepsilon.
> $$
> Since we obtain empirical estimates of $\lambda_0$ and $\lambda_1$, this
> expression provides a practical guideline for determining the scale of
> $T_{\mathrm{cut}}$, and thus can be implemented in practice. In our case in LQR Isotropic experiment, for example, we have
> $\lambda_1 - \lambda_0 = 2\sqrt{3}\,\beta$, so choosing
> $T - T_{\mathrm{cut}} = 1.8$ corresponds to $\varepsilon \approx 0.05$ in
> the above calculation.

---

> > ### Author Response · Authors · 2025-11-24
> >
> > Reply, part 2 out of 2:
> >
> > **Improvement compared to IDO and FBSDE.** Our approach exploits the specific structure of the problem: HJB equation admits a linear PDE reformualtion. The standard IDO and FBSDE schemes are designed to apply to general problem settings and therefore do not make use of this crucial property. In particular, this structure enables us to derive more accurate estimators for the control problem through explicit mathematical expressions. Leveraging this problem-specific information is precisely what yields the observed improvement in $L^2$ error compared with standard IDO and FBSDE methods.
> >
> > **Gradient drift** When the drift is not a gradient, there is currently no straightforward extension of our method. The core of our approach relies on Theorem 1, which requires the operator $\mathcal{L}$ to be self-adjoint. One possible direction would be to replace self-adjointness with a weaker condition, such as normality, and to work with a more general spectral theorem, but this lies beyond the scope of the present work. In addition, our algorithm explicitly uses the form $\mu\propto e^{-2\beta E}$ when computing inner products with respect to the invariant measure $\mu$. This representation is specific to the gradient-drift setting. Extending the method to non-gradient drifts would therefore require modifying this part of the algorithm to accommodate a more general invariant measure.

---

### Meta-Review · Area_Chair_9WYW · 2025-12-28

**Summary:**

This submission develops a principled spectral approach for long-horizon stochastic optimal control under the gradient-drift (linearly solvable) setting by showing unitary equivalence between the relevant generator and a Schrödinger operator with discrete spectrum, and by leveraging the leading eigensystem to represent long-horizon structure efficiently. It includes an analytic treatment for symmetric LQR via the quantum harmonic oscillator correspondence, and proposes a neural eigenfunction-learning pipeline with a loss reweighting tailored to the control objective. Across reviewers, the work is viewed as technically strong and clearly written, with particularly positive assessment of the theoretical development and the overall framing; empirical evidence supports substantial gains in long-horizon regimes relative to generic solvers, while the scope remains specialized to settings where the key structural assumptions hold. Overall, I recommend Accept (poster) as a solid, well-motivated contribution at the intersection of control/PDE theory and modern learning-based solvers, with clear value to the community even if some theoretical and empirical extensions are left for future work.

**Reviewer Concerns:**

Reviewer AKir’s main concerns focused on clarity and positioning of the core spectral result (Theorem 2), the adequacy of the appendix “proof,” practical selection of the cutoff time T_cut, and requests for deeper discussion of convergence/scaling and of the source of the reported accuracy/complexity improvements; the authors’ rebuttal directly clarifies that Theorem 2 is not claimed as novel, reframes the presentation accordingly, and explains that discreteness/no finite accumulation is a theoretical property rather than an empirical check, while also providing additional empirical evidence across dimensions and a concrete heuristic for setting T_cut based on the estimated spectral gap. Reviewer KCj8 raised questions about the practical breadth of the gradient-drift assumption, the non-asymptotic relevance of the correction term at early times, the relationship between the proposed loss variants and normalization, and potential instability/sensitivity of the relative loss; the rebuttal strengthens the motivation with examples where gradient drift arises, explains that the time-dependent correction term is learned explicitly in the procedure, and candidly distinguishes what is rigorously characterized (variational loss minimizers) from what is not yet fully proven (relative-loss minimizers and approximation-error bounds), while addressing implementation details for T_cut, eigenfunction interpretation, and the λ0 estimation protocol. Reviewer Y3as’s substantive concern about whether improvements are largely explained by an ergodic/turnpike approximation is addressed by an added “ergodic controller” comparison and discussion; the reviewer explicitly indicates that this resolves their main concern and that they did not change their review. Remaining limitations are primarily the restricted assumption set (gradient drift/self-adjoint structure), incomplete theory for the relative-loss minimizer characterization and error bounds, and the inherently non-convex nature of the learning dynamics that precludes clean convergence-rate guarantees; these are acknowledged appropriately and do not, in my view, outweigh the paper’s clear conceptual and methodological contribution.

**Reviewer Scores:**

Reviewer Y3as explicitly states that the rebuttal addressed their main concern and that they made no changes, so I estimate their score remains at 8. Reviewer KCj8’s questions are largely answered at the implementation/motivation level, while one theoretical point (full characterization of the relative-loss minimizers and associated bounds) is acknowledged as future work; I therefore estimate their score most likely remains at 6, with a plausible upward move to 7 if the added clarifications are weighted heavily. Reviewer AKir’s most consequential issues were about theorem positioning/clarity and practical guidance (T_cut and sources of improvement); because the rebuttal directly concedes and fixes the novelty/clarity point, provides a concrete T_cut heuristic, and adds empirical scaling evidence, I estimate a modest increase from 4 to 5 is likely, while recognizing that lack of formal convergence-rate theory could also justify a flat score in a strict reading. Under this post-rebuttal estimate, the overall profile supports an Accept (poster) recommendation.

---

### Decision · Program_Chairs · 2026-01-26

Accept (Poster)